# A long-term self-driven metronomic photodynamic system for cancer therapy

Weili Wang[1,2,3,4,5,6,9], Binglin Ye[1,2,3,4,5,6,9], Yao Liu[1,2,3,4,5,6], Zhi Li[7],
Qianying Huang[1,2,3,4,5,6], Jialin Zhou[1,2,3,4,5,6], Min Hu[1,2,3,4,5,6], Jun Jiang[1,2,3],
Weilin Wang[1,2,3,4,5,6] ✉, Zhengwei Mao [1,2,3,4,5,6,8] ✉ & Yuan Ding [1,2,3,4,5,6] ✉

Metronomic photodynamic therapy is a long-term, low-dose treatment strategy that employs optical devices with continuous photosensitizer administration and requires stable device attachment with a consistent power source. These factors significantly limit patient mobility. Currently, no metronomic photodynamic therapy modality can operate independent of external devices, underscoring the critical need for in vivo light sources that function without external energy inputs. In this study, we integrate self-luminous bacteria with a photosensitizer in alginate microcapsules to create a self-driven metronomic photodynamic therapy that can be securely implanted within a tumour, thereby enabling continuous light emission without requiring an external energy source or ongoing replenishment of photosensitive reactants. By harnessing nutrients from the tumour microenvironment, this system sustains the generation of reactive oxygen species. A single injection effectively eliminates larger tumours (>300 mm$^3$) in an opaque melanoma mouse model and transplanted hepatocarcinoma rabbit model. Self-driven metronomic photodynamic therapy demonstrates advantages over traditional photodynamic therapy, indicating its potential as a versatile therapeutic approach for cancer treatment with deeply situated lesions.

Recently, significant progress has been made in the development of light-based technologies for diagnosis, therapy, and surgical interventions[1]. Photodynamic therapy (PDT) is an efficacious and clinically validated treatment for cancers within the superficial layers of various organs, including the oesophagus, stomach, lung, and cervix[2–4]. In standard clinical practice, optical fibres channel light towards internal lesions, delivering short-term, high-intensity photoirradiation (lasting several tens of minutes at intensities >100 mW/cm$^2$)[5,6]. However, excessive light exposure can induce thermal damage to tissues, causing necrosis and inflammation that potentially compromise organ functionality[7]. Conversely, inadequate light exposure fails to produce the desired anti-tumour effects[8]. Thus, maintaining an optimal balance between minimising radiation-induced harm and effectively eliminating cancer cells remains a major challenge in PDT.

[1]Department of Hepatobiliary and Pancreatic Surgery, The Second Affiliated Hospital, Zhejiang University School of Medicine, Zhejiang, Hangzhou, P. R. China. [2]Key Laboratory of Precision Diagnosis and Treatment for Hepatobiliary and Pancreatic tumour of Zhejiang Province, Zhejiang, Hangzhou, P. R. China. [3]Research Center of Diagnosis and Treatment Technology for Hepatocellular Carcinoma of Zhejiang Province, Zhejiang, Hangzhou, P. R. China. [4]Center for Medical Research and Innovation in Digestive System Tumors, Ministry of Education, Zhejiang, Hangzhou, China. [5]ZJU-Pujian Research & Development Center of Medical Artificial Intelligence for Hepatobiliary and Pancreatic Disease, Zhejiang, Hangzhou, P. R. China. [6]Cancer Center, Zhejiang University, Zhejiang, Hangzhou, P. R. China. [7]Department of Interventional Radiology, the First Affiliated Hospital of Soochow University, Soochow University, Jiangsu, Suzhou, China. [8]MOE Key Laboratory of Macromolecular Synthesis and Functionalization Department of Polymer Science and Engineering, Zhejiang University, Zhejiang, Hangzhou, China. [9]These authors contributed equally: Weili Wang, Binglin Ye. ✉e-mail: wam@zju.edu.cn; zwmao@zju.edu.cn; dingyuan@zju.edu.cn

To improve both the safety and effectiveness of PDT, an method called metronomic photodynamic therapy (mPDT) has been developed[9]. This approach involves prolonged, low-intensity photo-irradiation sessions (ranging from a few hours to several tens of hours at intensities <10 mW/cm²), which markedly diminishes the likelihood of thermal damage while effectively eliminating cancer cells. Nevertheless, mPDT requires continuous administration of a photosensitizer (PS) and relies on an implantable PDT device powered by batteries[10] and wireless[11] technology, potentially complicating treatment. To date, no mPDT modalities that can function without external supporting devices exist, highlighting the urgent need for in vivo light sources that can operate independently of external energy sources.

Currently, bacteria-based bioactive drugs are receiving increasing attention for their tumour-targeting capabilities and roles in enhancing immune responses[12,13]. Drawing inspiration from the bioluminescent organs of deep-sea anglerfish, scientists discovered bioluminescent bacteria capable of sustaining light emission by utilising host nutrients regulated by quorum sensing[14]. Although these bacteria exhibit the potential for self-driven photodynamic treatment, maintaining the safety, stability, and functionality of bacteria-based optical devices in vivo poses a substantial challenge for the practical deployment of fully implantable mPDT systems[15]. Therefore, employing this strategy for internal tumour treatment requires stable fixation of bioluminescent bacteria and PS to enable continuous, localised light delivery and the generation of reactive oxygen species (ROS).

In this work, we combine bioluminescent bacteria and Neutral red (NR) photosensitizer[16] within alginate microcapsules (MCs)[17] to develop the self-driven metronomic photodynamic system (Sd-PDT) for cancer therapy. This system diverges from traditional mechanical- and chemical-driven PDT methods by enabling continuous, uniform light emission without external energy support or the continuous replenishment of photosensitive reactants, and produces prolonged, low-dose light emissions, sustaining continuous emission for approximately 50 h. Grafting NR onto the surface of bacteria@alginate microcapsules (PB@MCs) significantly prolongs self-supported ROS generation. The effectiveness of PB@MCs is evaluated in four cancer cell lines and two animal tumour models[18]. A single injection of PB@MCs is sufficient to effectively eradicate large tumours within 2 days. Additionally, our results demonstrate that Sd-PDT through PB@MCs triggers a strong antitumour immune response, effectively inhibiting tumour metastasis and recurrence. This study highlights the potential of Sd-PDT as a therapeutic strategy applicable to certain tumours, offering promising clinical prospects.

## Results
### Engineering and characterisation of PB@MCs
Initially, our experimental approach involved constructing the anticipated PB@MCs by subjecting bioluminescent bacteria to physiological conditions, encapsulating them within MCs, and chemically grafting PS molecules onto the surfaces of the MCs (Fig. 1A). We selected the bioluminescent bacteria *Vibrio harveyi BB170* (V.H.BB170), *Aliivibrio fischeri-bio115653* (F.A.115653) and *Aliivibrio fischeri-7744* (F.A.7744) (Supplementary Fig. 1), which are typically cultured in 2216E medium at 25 °C. These three bacterial strains were evaluated for viability and bioluminescent activity under physiological conditions. Acclimatisation was gauged by measuring the bacterial density using the optical density (OD) at 600 nm (Supplementary Fig. 2). Despite the pronounced suppression of growth and bioluminescence in F.A.115653 and F.A.7744 at elevated temperatures from 25 °C to 37 °C, V.H.BB170 demonstrated adaptability, maintaining standard growth and luminescent output. Consequently, V.H.BB170 bacteria were selected for the further development of PB@MCs (Fig. 1B and Supplementary Fig. 2). For photosensitisation and initiation of ROS generation, N-hydroxysuccinimide neutral red (NHS-NR) and N-hydroxysuccinimide chlorin e6 (NHS-Ce6) photosensitizer are both commonly used PS molecules. Since the

V.H.BB170 strain emits robust bioluminescence within the 400–600 nm range, we compared the UV–Vis absorption spectra of these two PS molecules to identify the one best matching the bacterial emission range. As shown in Supplementary Fig. 3, the emission of V.H.BB170 ideally aligns with the absorption range of NHS-NR. Additionally, NR functions as a Type I photosensitizer, primarily generating hydroxyl radicals (•OH)[19,20], while Ce6 acts as a Type II photosensitizer, mainly generating singlet oxygen ($^1O_2$). The generation of •OH is less dependent of oxygen availability, suggesting that NR, as a Type I photosensitizer, may offer greater advantages in hypoxic tumour regions.

To develop a durable and persistent Sd-PDT system in vivo, an ionically crosslinked alginate hydrogel was selected as the matrix owing to its well-known biocompatibility with cell culture[17,21]. PB@MCs were engineered by encapsulating bioluminescent bacteria within MCs via electrostatic attraction using an electrostatic droplet generation system. PS molecules were chemically grafted onto the surface of the MCs as follows (Fig. 1A): First, the sodium alginate hydrogel precursor was blended with V.H.BB170 bacteria to achieve a uniform mixture. Second, the solution was dispersed into consistent droplets under an electrostatic attraction, and the droplets were immersed in a $CaCl_2$ solution. $Ca^{2+}$ ions permeated the droplets, effectuating ionic cross-linking of alginate chains and encapsulation of the bacterial cells within the hydrogel matrix (denoted as the B@MC composite). Third, the alginate-calcium microspheres were coated with poly-L-lysine (PLL), a water-soluble polycation that is resistant to enzymatic breakdown and prevents microbial escape[17] (Fig. 1C and Supplementary Figs. 4 and 5). The morphology and size of the B@MCs prepared by an electrostatic attraction were controlled to be within 150 μm, making this small size particularly suitable for engineering biocompatible Sd-PDT for internal tumours. These PLL-B@MCs were further cultured in 2216E medium at 37 °C, allowing the bacteria to proliferate to the platform stage (Fig. 1D). Fourth, PLL, which comprises abundant amino functional groups ($-NH_3$) enabled subsequent chemical conjugation. The reaction between PLL and NHS-NR formed covalent amide linkages via the active NHS ester, which interacted specifically with the $-NH_3$ group. Subsequently, the NR was efficiently conjugated to the MCs, resulting in the formation of PB@MCs. The bacterial density and PS (NHS-NR) binding affinity of each PB@MC were thoroughly assessed by plotting the linear relationship between bacterial OD and CFU/mL (Supplementary Fig. 6) and measuring the absorbance of unreacted NHS-NR at 452 nm after collecting the resultant solution (Supplementary Fig. 7). Upon calculation, the formula for the average of PB@MC ($1 \times 10^3$ bacteria cells and 0.028 μg of PS per MC) was established.

Confocal laser scanning microscopy (CLSM) was used to examine PB@MCs. Notably, while empty MCs displayed no bioluminescence, microspheres containing bioluminescent bacteria, specifically B@MCs and PB@MCs, emitted vivid blue light. This demonstrated that the PS coating on the surfaces of the MCs did not affect the light emission ability of the bacteria and that the bacteria were capable of independent light emission without external excitation while efficiently converting nutrients in the culture medium into light (Fig. 1E). These results confirmed that the encapsulated bioluminescent bacteria maintained their light emission characteristics. Confocal microscopy analysis demonstrated that bacterial cells within the MCs self-assembled into structurally stable aggregates while maintaining sustained bioluminescent activity, with no detectable bacterial bioluminescence signals outside the MCs (Fig. 1E). Furthermore, we evaluated the emission spectra of B@MCs before and after PS modification. Predictably, both groups of encapsulated bioluminescent bacteria exhibited a Gaussian emission profile centred at ~480 nm with a full width at half-maximum of 100 nm (Fig. 1F).

In practical applications, PB@MCs can be deployed as long-term Sd-PDT devices in tumour microenvironments. However, maintaining long-lasting light emission poses a challenge in vitro because persistent luminescence requires continuous nutrient availability. Once

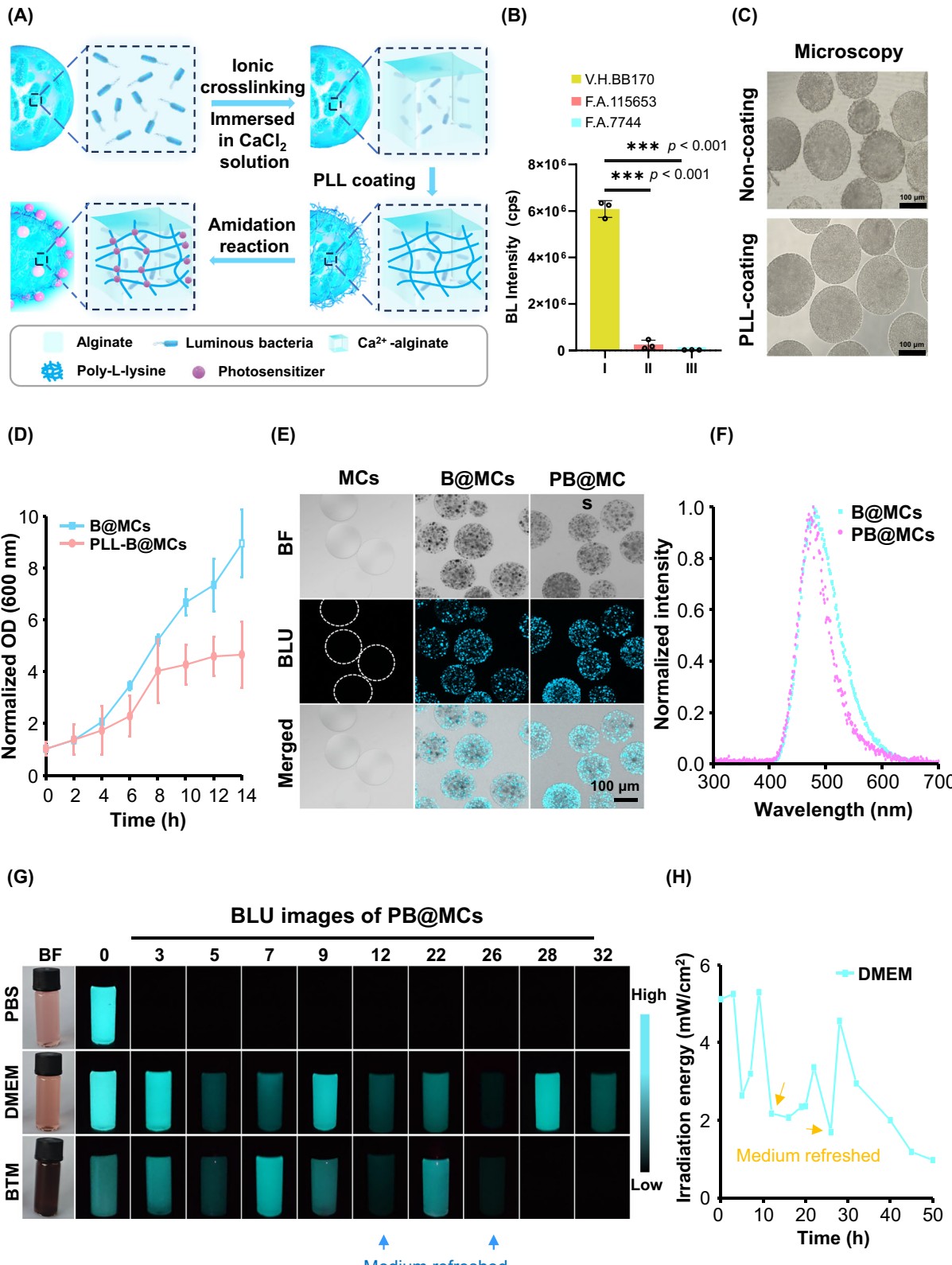

nutrients in the culture medium are depleted, bacterial luminescence decreases[22]. To mimic the nutritional environment within actual tumours, in vitro experiments were conducted with frequent replenishment of the culture medium to simulate a continuous nutrient supply. PB@MCs were evaluated for their long-term bioluminescence capabilities in physiological solutions and various medium formulations, including Dulbecco's modified Eagle medium (DMEM) and B16

tumour homogenate (BTM). As shown in Fig. 1G, when bioluminescence diminished, immediate supplementation of fresh culture medium revitalised the light emission of PB@MCs, which was maintained for 32 h in transparent bottles. A laser power meter was used to plot the linear relationship between the bioluminescence intensity ($10^4$ cps) and the power density (mW/cm$^2$) of PB@MCs (Supplementary Fig. 8), and further calculations revealed that PB@MCs can emit

**Fig. 1 | Fabrication, demonstration, viability, stability, and self-luminescence duration of PB@MC living composites. A** Diagram outlining PB@MC construction. **B** Bioluminescence (BL) intensity of three types of bioluminescent bacteria at 37 °C ($n = 3$ independent experiments). The data are presented as mean ± SD. Differences were considered statistically significant at ***$p < 0.001$, compared to the data for the F.A. 115653 and F.A. 7744 groups according to one-way ANOVA test combination with Tukey's multiple comparison test performed using GraphPad Prism 9 XML project software. **C** Representative images of B@MCs both with and without PLL coating. Coated and uncoated PLL-B@MCs were cultured in 2216E medium for 8 h before collection for microscopy. **D** Growth curves for B@MCs and PLL-B@MCs cultured in 2216E medium at 37 °C for 14 h. Microcapsules (50 μL) were gathered within 96-well transparent plates, and their optical density at 600 nm was measured using a microplate reader ($n = 6$ independent experiments). The data are presented as mean ± SD. **E** Confocal imaging of PB@MCs. Empty MCs, MCs containing bacterial cells (B@MCs), and fully constructed PB@MCs were imaged for BL using confocal microscopy with a 400–500 nm channel. **F** BL emission spectrum of B@MCs and PB@MCs. 500 μL samples were cultured in equal volumes of DMEM to analyse BL emission spectra from 300 to 700 nm using an Edinburgh FS5 spectrophotometer. The spectra were analysed using Gaussian function fitting on OriginLab 8.0 software. **G** BL images of PB@MCs in various media. PB@MCs were cultured in PBS, DMEM, and B16 tumour homogenate (BTM) in transparent penicillin bottles at 37 °C for 32 h in vitro. Photos were taken with a 3.2 s exposure. Medium was refreshed at 12 and 26 h (blue arrow). **H** In vitro experiments to investigate persistent irradiation energy (mW/cm²) capabilities of PB@MCs. PB@MCs (100 μL, $3.6 \times 10^4$/mL) were cultured in DMEM medium in 96-well black plates at 37 °C. BL intensity was measured at 450 nm for 50 h, medium was refreshed at 12 and 26 h (orange arrow). Source data are provided as a Source data file.

bioluminescence at a power density of 1–5 mW/cm² in DMEM medium for up to 50 h (Fig. 1H), which is suitable for the mPDT photoirradiation dose[23] and significantly surpasses the duration of all currently available chemical-driven PDTs[14].

## Anti-tumour cells and ROS generation test of PB@MCs

We explored the efficacy of Sd-PDT using PB@MCs on cancer cells. This study assessed the effect on four distinct cancer cell lines, two liver cell lines (Hep3B and VX2) and two melanoma cell lines (A375 and B16). The 24-Transwell and 6-Transwell setups prevented direct physical contact between the PB@MCs and cancer cells, facilitating the assessment of the Sd-PDT effects. PB@MCs and cancer cells were placed in the upper chamber and the lower chamber, respectively (Fig. 2A). Cell viability was assessed using the LIVE/DEAD staining assay (Calcein/PI Assay Kit) and visualised using CLSM. After 8 h, a significant red signal increase in cell mortality was observed in all cell lines in the PB@MC group, suggesting a potent cytotoxic effect (Fig. 2B and Supplementary Fig. 9). In contrast, the cells treated with B@MCs and P@MCs, which contained only bacteria or PS, maintained over 90% cell viability, indicating their lower toxicity. Cellular metabolic functions were further analysed using a colorimetric CCK-8 assay (Fig. 2C and Supplementary Fig. 10) and aligned with the results from LIVE/DEAD staining (Calcein/PI Assay Kit), demonstrating significant cytotoxic effects in the PB@MC groups across all tested cancer cell lines. Apoptosis induced by PB@MCs was investigated using flow cytometry (Fig. 2D, E and Supplementary Fig. 11), Annexin V-FITC(+) and DAPI(−) cell populations are defined as early apoptotic cells, while Annexin-FITC(+) and DAPI(+) cell populations are defined as late apoptotic cells. Toposide VP16[24], a typical anti-tumour drug, was used as a positive control. The proportions of cells exhibiting late apoptosis (DAPI fluorescence signal) were 2.67%, 9.19%, 19.5% and 43.3% in the Control, P@MCs, B@MCs and PB@MCs groups, respectively, demonstrating that cytotoxic effects were predominantly observed in cells treated with PB@MCs.

To verify that these anti-tumour effects were predominantly attributable to ROS production, we analysed the ROS generation capabilities of PB@MCs both in vitro (physiological solutions) and in vivo (tumour environments). For in vitro testing, PB@MCs were incubated in 24-well Transwell at 37 °C for 2 h, with no treatment (vehicle control), and B@MCs and P@MCs as control groups. The ROS levels in Hep3B and A375 cells were assessed using 2′,7′-dichlorodihydrofluorescein diacetate (DCFH-DA) staining[25], which measures ROS activity by detecting the oxidation of nonfluorescent DCFH to fluorescent 2′,7′-dichlorodihydrofluorescein (DCF) (Fig. 3A and Supplementary Fig. 12). In contrast to the limited signals from the cells treated with B@MCs, intense fluorescence was observed in the cells treated with PB@MCs, indicating more ROS generation. We also evaluated the bioluminescence performance and duration of PB@MCs in mice. As shown in Supplementary Fig. 13, PB@MCs exhibited bioluminescent signals in tumours for at least 10 h, confirming their bioluminescent capability within hypoxic tumour microenvironments.

However, the bioluminescent signals of PB@MCs gradually attenuated over time, with signal weakening initiating from the tumour core and progressively diminishing toward the periphery. This phenomenon may be attributed to the hypoxic gradient within tumours, where oxygen deprivation intensifies toward the core while relatively oxygenated conditions prevail in the peripheral regions. Consequently, stronger self-luminescent signals were observed in areas distal to the tumour centre. The in vivo bioluminescence duration of PB@MCs was shorter than the 50-h duration observed in vitro, possibly because the signal intensity fell below the IVIS system's detection limit (potentially influenced by melanin absorption in melanoma and limited tissue penetration depth), or more likely due to ongoing oxygen consumption during sustained bacterial bioluminescence within tumours. Following the experiment, tumours were aseptically harvested, weighed, and processed according to the Tissue •OH assay kit (BBoxiProbe O28 probes)[26] to evaluate the in vivo •OH levels. As shown in Fig. 3B, mice treated with PB@MCs exhibited significantly higher fluorescence intensity compared to the other groups, demonstrating enhanced •OH generation and confirming the in vitro findings. To verify that the cumulative ROS production depends on the bioluminescence intensity and exposure duration of PB@MCs, ROS generation was continuously monitored for approximately 50 h. The duration and intensity of ROS generated by Sd-PDT were compared with those of conventional PDT (external LED at 300 mW/cm² irradiating NR for 1 h per session). For the Sd-PDT group, PB@MC microspheres loaded with an equivalent NR dose were used. ROS fluorescence intensity was quantified using the DCFH probe at each time point. As shown in Fig. 3C, the LED group exhibited a significant increase in ROS fluorescence signals during 1-h irradiation, indicating rapid and strong initial ROS generation. However, ROS production ceased immediately after irradiation ended. Conversely, the PB@MC group displayed persistent ROS generation throughout the treatment session (~50 h), despite lower initial fluorescence intensity compared to the LED group. These results confirm the sustained ROS production of the Sd-PDT system in a single treatment session, reflecting their persistent effectiveness in the tumour environment. The cumulative ROS quantity in the PB@MC group significantly exceeded that of the LED group (Fig. 3D). These findings provide substantial evidence of the durable ROS-generating capacity of PB@MCs.

## Metabolism analysis of tumours after PB@MCs treatment

In addition to the effects of ROS, metabolomic analyses were performed to elucidate the anti-tumour mechanisms of PB@MCs. Melanoma tumour-bearing mice treated with PB@MCs, B@MCs, PS and PS with LED radiation were further investigated to decipher the intrinsic mechanisms through which PB@MCs impede tumour progression utilising metabolomic detection and analysis, focusing on specific pathways of metabolic reorganisation within the tumour microenvironment post-treatment[27,28]. Using non-targeted metabolomics, the levels of various metabolites in the tumour microenvironment after PB@MC treatment

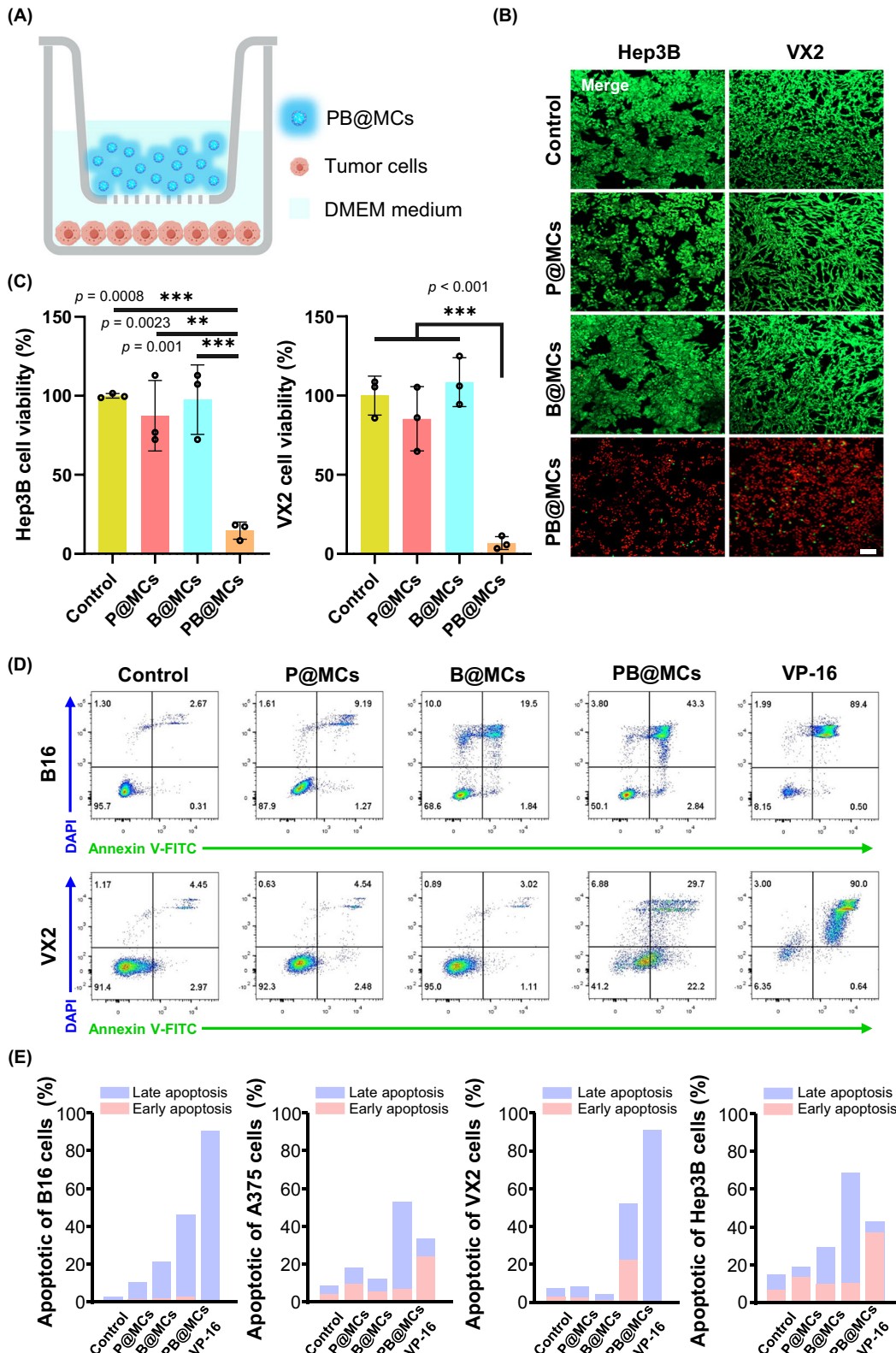

were compared with those in the control groups. Principal component analysis highlighted the metabolite distribution profiles across different treatments (Fig. 4A). PB@MCs induced significant alterations in the metabolic profile, markedly distinguishing these profiles from those of the other groups, indicating the potential metabolic reorganisation induced by PB@MCs. As shown in Fig. 4B, a Venn diagram revealed that the metabolite composition among the groups was similar, suggesting

that PB@MCs may inhibit tumour progression by suppressing the expression of components within the same metabolic pathways. Supporting this hypothesis, volcano plot analysis (Fig. 4C) identified 90 significantly downregulated and 50 upregulated components in the PB@MC group compared to those in the control group (fold change (FC) > 2 or <0.5 and $p < 0.05$). We further performed cluster analysis of the metabolites derived from metabolic profiling. As shown in Fig. 4D,

**Fig. 2 | Anti-tumour efficacy of PB@MC living composites. A** Diagram depicting the anti-tumour action of PB@MCs using a Transwell model. Images are created by figdraw.com. **B** Fluorescence microscopy images following LIVE/DEAD staining (Calcein/PI Assay Kit) in vitro. Hep3B and VX2 cells were treated with 200 μL of either PBS, Empty MCs linked to photosensitiser (P@MCs, contain 0.22 μmol NR), MCs encapsulating V.H.BB170 bacteria (B@MCs), or PB@MCs for 8 h, followed by staining with a LIVE/DEAD kit for microscopy visualisation. Red fluorescence represents dead cells and green fluorescence represents live cells. Scale bars: 100 μm. **C** In vitro cell viability assessment. Hep3B and VX2 cells were exposed to 200 μL of P@MCs, B@MCs, or PB@MCs in dark conditions for 8 h. Viabilities were determined using an CCK-8 assay ($n = 3$ independent experiments). The data are presented as mean ± SD. Statistical significance is noted with ***$p < 0.001$ and **$p < 0.01$, compared to the data for P@MC, B@MC and control group according to one-way ANOVA with Tukey's multiple comparisons performed using GraphPad Prism 9 XML project software. **D** In vitro test of apoptotic cell detection via flow cytometry. Representative images and **E** quantitative analysis. B16, A375, VX2 and Hep3B cells ($5 \times 10^5$ cells/well) in 6-Transwell plates were treated with bare B@MCs, P@MCs (contain 0.44 μmol NR), and PB@MCs at concentrations of 400 μL or 400 μg/mL VP-16. Following 8 h of incubation, Annexin V-FITC(+) and DAPI(−) cells are defined as early apoptotic cells, whereas Annexin-FITC(+) and DAPI(+) cells are defined as late apoptotic cells for analysis by flow cytometry. Source data are provided as a Source data file.

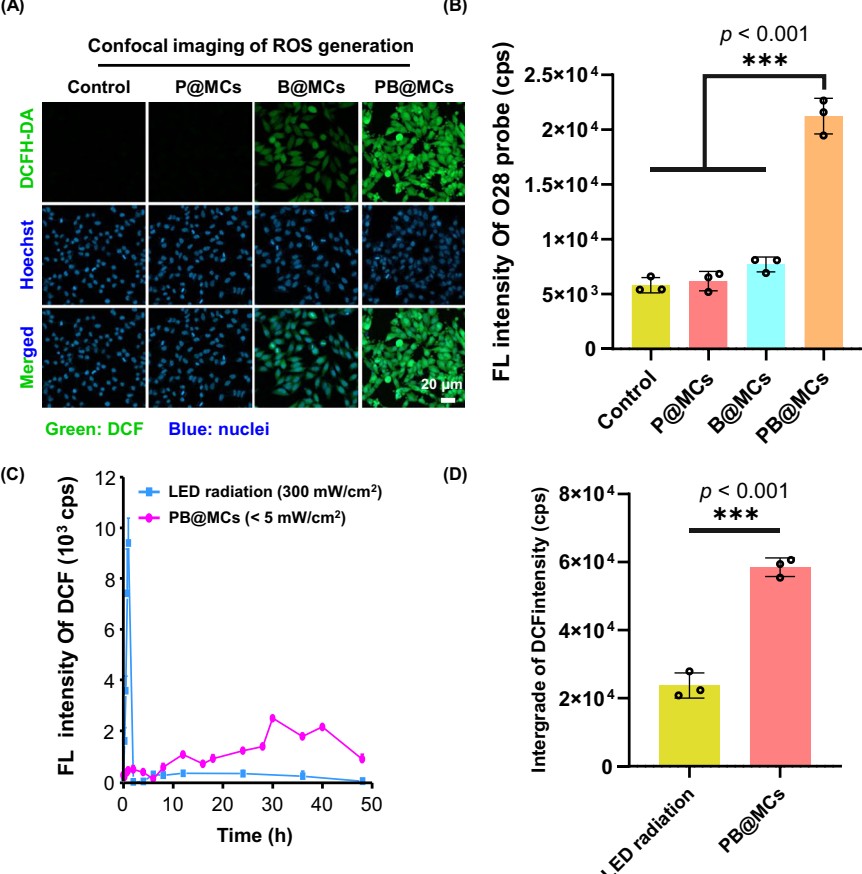

**Fig. 3 | Characterisation of ROS generation efficiency in vitro and in vivo by PB@MCs. A** Confocal microscopy of Hep3B cells using DCFH-DA probe in vitro. Hep3B cells, seeded in 24-Transwell plates at $5 \times 10^4$ cells/well, were treated with 100 μL P@MCs, B@MCs and PB@MCs and subsequently stained with the ROS-specific probe DCFH-DA after 2 h. **B** In vivo detection of ROS content. ROS production was assessed using the Tissue Hydroxyl Radicals Assay Kit (BBoxiProbe O28 probe). For comparison, B16 tumour-bearing mice were treated with either 50 μL PBS (control group), empty MCs linked to photosensitiser (P@MCs, contain 0.055 μmol NR), MCs encapsulating V.H.BB170 bacteria (B@MCs), or PB@MCs for 10 h; 190 μL of B16 tumour homogenate (50 mg/mL) and 10 μL of BBoxiProbe O28 working solution were added to a 96 black well plate, the plate was incubated at 37 °C in darkness for 20 min, and fluorescence intensity was detected using a microplate reader (excitation at 488 nm, emission at 520 nm) ($n = 3$ independent experiments). The data are presented as mean ± SD. Statistical significance is noted with ***$p < 0.001$, compared to the data for the P@MC, B@MC and control group according to one-way ANOVA with Tukey's multiple comparisons performed using GraphPad Prism 9 XML project software. **C** ROS generation duration test in vitro by PB@MCs (30 μL, $3.6 \times 10^4$/mL). Fluorescence intensity of DCF was monitored ~50 h using a microplate reader ($n = 3$ independent experiments). The LED radiation group (300 mW/cm² for 60 min LED radiation of with an equivalent dose of NHS-NR, 0.32 mM in 100 μL medium) was used for comparison. **D** Quantitative analysis of ROS generation after a single treatment of PB@MCs and LED group ($n = 3$ independent experiments). The data are presented as mean ± SD. Statistical significance is noted with **$p < 0.001$, compared to the LED radiation group according to two tailed $t$-test using GraphPad Prism 9 XML project software. Source data are provided as a Source data file.

metabolite expression levels from tumour tissues under different treatments were standardized using Z-score normalization. The top 30 metabolites ranked by relative abundance were selected for the clustered heatmap. The results demonstrate significant differences in metabolite expression between the PB@MC-treated group and the other groups. PB@MCs significantly reduced the metabolites involved in critical pathways such as glucose and lipid metabolism, including glucose 6-

phosphate, phosphatidylethanolamine (Pe), glyceraldehyde and phosphatidylcholine (Pc). Glucose-6-phosphate plays a central role in carbon metabolism by linking glycolysis to the pentose–phosphate pathway. Pc and Pe are the key components of eukaryotic cell membranes. The decrease in these metabolites suggests that PB@MC treatment reduced the utilisation of carbon sources by tumour cells and the synthesis of NADPH and nucleic acid-5-phosphate. This disrupts the redox balance in

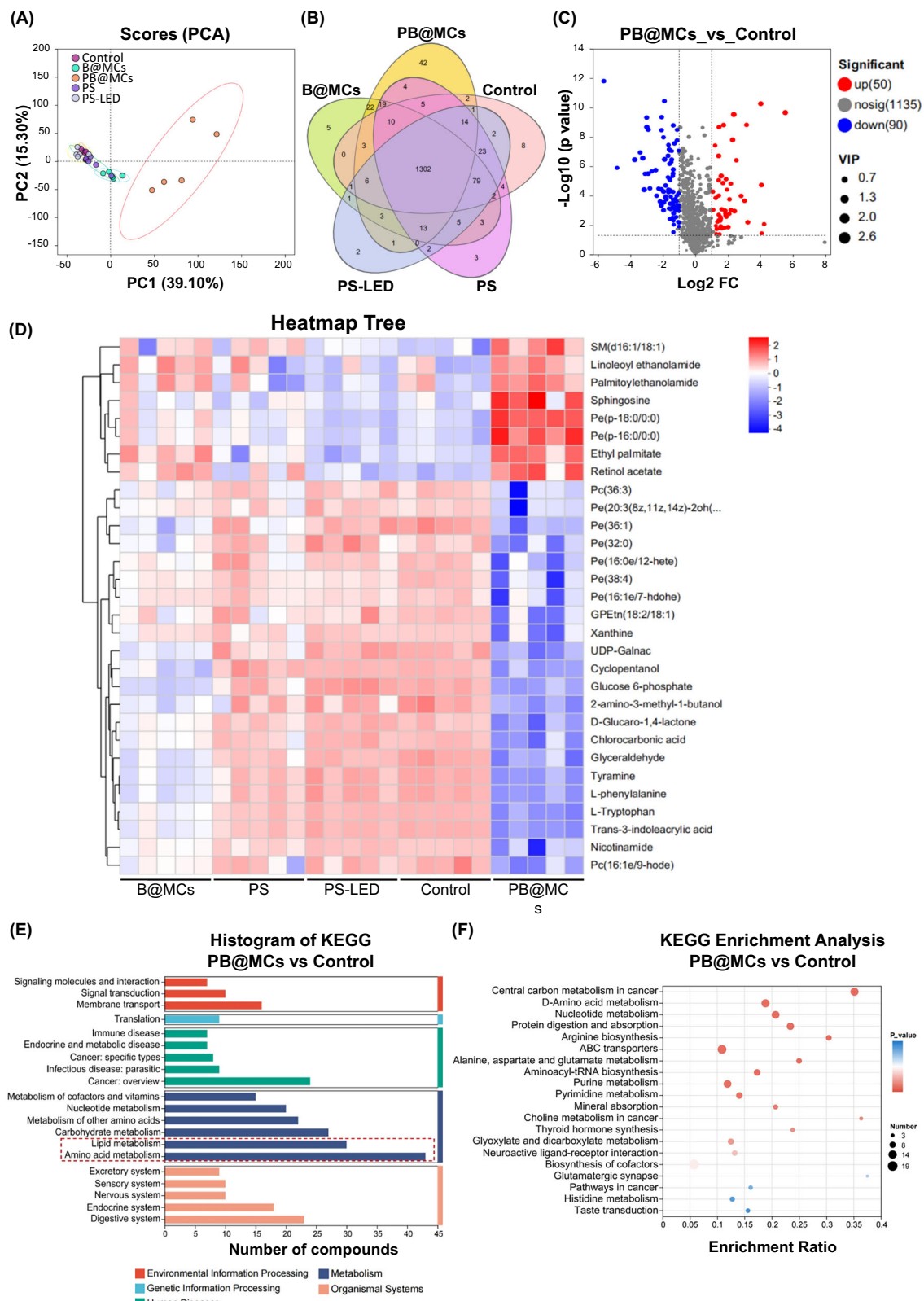

tumour cells, leading to high levels of ROS and impeding the synthesis of macromolecules such as nucleotides and glycerolipids that interfere with tumour cell proliferation. Pathway enrichment analysis of differential metabolites using the Kyoto Encyclopedia of Genes and Genomes (KEGG) database revealed a strong concentration of lipid and amino acid metabolism pathways (Fig. 4E). Further analysis revealed that metabolic pathways closely associated with glutathione metabolism, including

cysteine and methionine metabolism, biosynthesis of cofactors, ABC transporters, and thyroid hormone synthesis, were significantly suppressed following PB@MC treatment, contributing to increased ROS stress and antitumour effects (Fig. 4F and Supplementary Fig. 14). Additionally, phospholipid biosynthesis pathways, including glycerolipid and sphingolipid metabolism pathways, were notably downregulated. Because glycerolipids and sphingolipids are crucial

**Fig. 4 | Metabolism analysis of tumours after PB@MCs treatment. A** Principal component analysis. Orange ellipses indicate PB@MCs, and other colours indicate control groups. The metabolic profiles of the PB@MC and control groups were highly distinct. **B** Venn diagram of metabolites of PB@MCs, B@MCs, PS, PS-LED and Control groups ($n = 5$ independent experiments). **C** Volcano plot of metabolites. Down and upregulated metabolites are marked by blue and red dots, respectively, and the grey dots represent metabolites with no significant difference. The vertical dotted line denotes |log2(fold change)| = 1, and the horizontal dotted line shows -lg($p$ value) = −lg(0.05). $p$ values are determined by two-sided student's $t$-test. **D** Cluster analysis of the top 30 differential metabolites. Each column

represents a sample, and each row represents a metabolite. The heat map value shows the relative expression level ($n = 5$ independent experiments). **E** KEGG compound analysis. Bar colour distinguishes the category of the pathways. **F** KEGG enrichment analysis of metabolites. −lg($p$ value) > −lg(0.05) is considered significant. Each animal was administered a single intratumoural injection. After 24 h, tumour samples were then stored in liquid nitrogen for subsequent metabolomics analysis. $p$-values are determined by two-sided Student's $t$-test and made for Benjamini and Hochberg FDR (BH) adjustments. The metabolism analysis data reported in this paper have been deposited on Zenodo Dataverse (DOI 10.5281/zenodo.16792981).

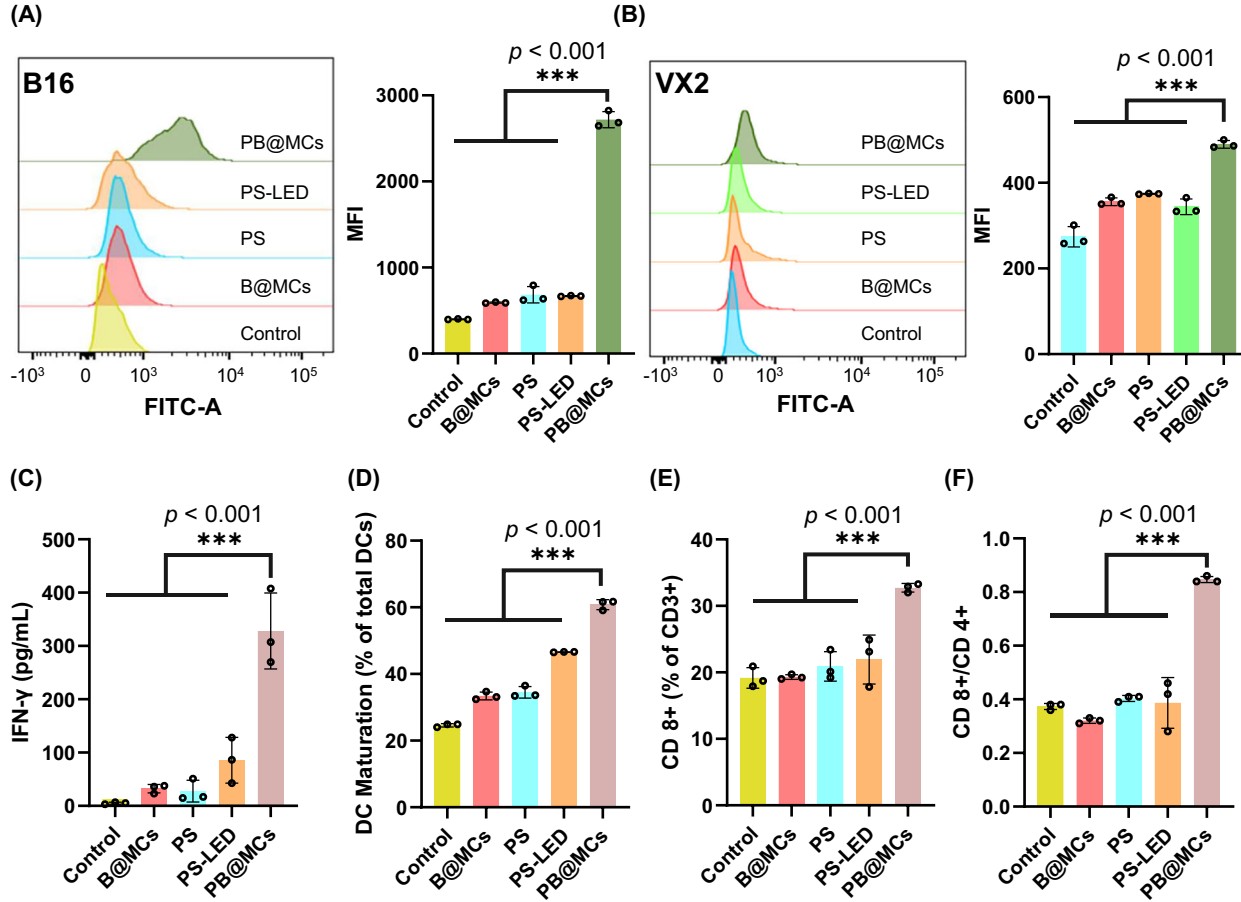

**Fig. 5 | In vitro *and* in vivo demonstration of immune response by PB@MCs.**
**A** CRT expression levels in B16 and **B** VX2 cells after various treatments as analysed by flow cytometry in vitro. After 6 h post-treatment, flow cytometry was used to quantify the mean fluorescence intensities for the PB@MC, a PS (NHS-NR, 0.033 μmol, 30 μL), PS with 450 nm LED light radiation (300 mW/cm²) (PS-LED, 0.033 μmol, 30 μL), B@MCs, and control groups ($n = 3$ independent experiments). **C** IFN-γ secretion levels in C57B6 mouse blood serum following various treatments for 3d ($n = 3$ independent experiments). **D** Percentage of DC maturation (CD80⁺CD86⁺ DCs out of total CD11c⁺

DCs) in B16 cells after exposure to different treatments in vitro ($n = 3$ independent experiments). **E** Quantitative analysis of CD8⁺ T cells (% of total CD3 + T cells) in B16 tumour tissues ($n = 3$ independent experiments). **F** CD8⁺/CD4⁺ ratio within tumour tissues ($n = 3$ independent experiments). Error bars, mean ± SD. Statistical significance is noted with ***$p < 0.001$, compared to the data for PS-LED, PS, B@MCs and control group according to one-way ANOVA with Tukey's multiple comparisons test performed using GraphPad Prism 9 XML project software. Source data are provided as a Source data file.

for maintaining cell membrane integrity, their downregulation limits the supply of the essential membrane components required for tumour cell expansion and repair, thereby enhancing ROS-mediated cytotoxicity. Moreover, pathways related to energy and nucleotide metabolism, such as oxidative phosphorylation, citric acid cycle, glutamate metabolism, and purine–pyrimidine metabolism, were markedly downregulated, further inhibiting the energy supply and macromolecule production necessary for tumour cell proliferation. The significant metabolic changes induced by PB@MCs suggest that their anti-tumour effects are mediated by the induction of oxidative stress, inhibition of energy metabolism, and suppression of synthesis and repair mechanisms in tumour cells.

## Immunology analysis of PB@MCs in tumour

PB@MC-induced Sd-PDT can cause effective immunogenic cell death (ICD) in cancer cells, promoting the expression of damage-associated molecular patterns (DAMPs)[29]. This process enhances the maturation and recruitment of DCs[30] to the tumour microenvironment. Subsequently, mature DCs activate cytotoxic T lymphocytes, thereby intensifying the anti-tumour immune responses. Among the critical DAMPs for ICD, calreticulin (CRT) is particularly significant for DC activation. As illustrated in Fig. 5A, B16 cells treated with PB@MCs, B@MCs, PS and PS-LED displayed CRT translocation from the endoplasmic reticulum to the cell membrane, with expression levels 6-, 1.35-, 1.85- and 1.8-fold higher than those observed in the control group. A similar pattern was observed in

the VX2 cells (Fig. 5B), where the CRT levels were 1.85-, 1.33-, 1.4- and 1.25-fold higher than those in the control groups. The concentration of interferon-γ (IFN-γ), a cytokine secreted by T cells, was measured in mouse serum using an enzyme-linked immunosorbent assay. Treatment with PB@MCs led to a pronounced increase in IFN-γ levels, indicating significant immune activation (Fig. 5C). Additionally, the maturation of DCs, marked by increased expression of CD80 and CD86, was quantified using flow cytometry. Compared to the control group, the proportion of mature DCs in the PB@MC-treated group significantly increased from 25% to 60% in vitro (Fig. 5D and Supplementary Fig. 15). These results indicate that PB@MCs effectively modified the immune microenvironment to suppress tumour growth. Encouraged by the promising responses observed in vitro, we further explored the in vivo effects of this immunotherapy. Intratumoural infiltration of T lymphocytes was assessed using flow cytometry. The PB@MC group showed a substantial increase in the presence of T cells within the tumour, particularly CD8$^+$ T cells, which reached ~33% of total CD3$^+$ cells in tumours treated with PB@MCs, which was significantly higher than that in other groups ($p < 0.001$) (Fig. 5E and Supplementary Fig. 16). Consequently, the CD8$^+$/CD4$^+$ ratio increased (Fig. 5F).

### Photodynamic therapeutic effects of PB@MC treatment in vivo

To validate the effect of PB@MC treatment in vivo, a melanoma syngeneic tumour model was created in mice via the subcutaneous introduction of B16 cells. When the tumours had grown to approximately 300 mm$^3$ (approximately 18 days), the mice were randomly allocated to one of five distinct therapeutic groups: no treatment (vehicle control), PB@MCs, B@MCs, PS treatment, or PS with LED irradiation (60 min, 300 mW/cm$^2$). Figure 6A shows the treatment regimen. Following a singular injection on day 0, the tumour size and body mass were monitored daily. Observations revealed rapid tumour enlargement in the untreated group, whereas PS and PS-LED treatments had negligible effects on melanoma progression. Conversely, the B@MC group experienced a recovery in tumour growth from day 4–6 onwards. Significantly, the volume metrics substantiated that PB@MCs provided the most pronounced antitumour response, leading to complete tumour eradication (Fig. 6B–D). In terms of tolerance, mice treated with PB@MCs showed no notable weight fluctuations (Fig. 6E). Survival analysis indicated that all mice except those treated with PB@MCs succumbed or died within 14 days. In stark contrast, PB@MCs not only completely treated the condition within 4 days but also notably extended survival to two months, with nine out of ten mice living beyond 60 days (Fig. 6F, G and Supplementary Fig. 17). These results suggest that PB@MCs are an effective Sd-PDT for melanoma in murine models. For biosafety evaluation, PB@MCs were administered to healthy mice intraperitoneally or subcutaneously. The blood test analysis performed on days 1, 3 and 7 revealed no significant abnormalities in WBC levels across all groups within the 7 day period. Serum procalcitonin (PCT) levels measured on days 1, 3 and 7 showed that compared to the control group, the PB@MC group exhibited elevated PCT on day 1. PCT levels in all groups decreased by days 3 and 7. These findings suggest that PB@MCs may induce a degree of inflammation, likely attributable to their generation of ROS, but this response gradually subsides after 3 days (Supplementary Fig. 18). Tissue samples were also collected for histological examination. Haematoxylin and eosin staining confirmed the absence of organ-specific pathological changes (Supplementary Figs. 19–21). Recognising the critical role of bacterial immune adjuvants in initiating immune activation[31], we postulated that PB@MCs could also amplify immunotherapy by stimulating adaptive immune activity within the tumour microenvironment. This was substantiated by immunostaining for TUNEL, CD4, CD8, CD3, CD206, and CD86 in a melanoma syngeneic tumour model. As shown in Fig. 6H–J, PB@MC treatment significantly ameliorated CD4$^+$ and CD8$^+$ production, enhanced CD3 expression, and reduced CD206 expression in tumour sections, indicating the activation of T cells and reduced M2-like macrophage infiltration within the tumour microenvironment[31,32].

### Effects of PB@MC implants on hepatocarcinoma

We further investigated the in vivo effects of Sd-PDT on hepatocarcinoma via direct implantation of PB@MCs into rabbit livers. Based on the outlined treatment protocol (Fig. 7A), rabbits afflicted with hepatocarcinoma were administered a single intratumoural injection of PB@MCs and monitored for a duration of 70 days. The progression of tumour growth was monitored using CT imaging during this period (Fig. 7B). Given the propensity of hepatocarcinoma to metastasise to the lungs, we assessed the lungs via CT imaging. Significant metastatic activity was evident in the lungs of untreated rabbits by day 14; extensive formation of tumour foci was observed by day 28. In contrast, the PB@MC treatment group exhibited no detectable lung metastases on day 70 (Fig. 7C, D). Oxaliplatin, a typical anti-tumour drug, was used as a clinical drug control. While tumour volumes in the oxaliplatin-treated rabbits increased to 3.8 cm$^3$ by day 14 and to 11.4 cm$^3$ by day 28 (Fig. 7E), the administration of PB@MCs arrested tumour expansion, and tumour dimensions reduced to indiscernible small cystic formations by the 14th day. Subsequent haematoxylin and eosin (H&E) staining confirmed the presence of swollen liver cells, necrotic liver cells, and fibrotic tissue at the periphery, whereas normal liver cells were present externally (Fig. 7F). Survival analysis using Kaplan–Meier plots demonstrated a significant survival benefit for the PB@MC-treated rabbits compared to controls (Fig. 7G). CD3$^+$ immunofluorescence and TUNEL staining of tumour tissues from various groups showed that PB@MC treatment induced strong antitumour immunity (Fig. 7H). H&E staining was performed to evaluate the histopathological features of hepatocarcinoma tumours and their pulmonary metastases. (Fig. 7I). These results demonstrate that PB@MC treatment markedly curtailed tumour growth, effectively inhibiting tumour metastasis and recurrence, and boosted survival rates in rabbits with hepatocarcinoma.

Recently, various PDT methodologies have been developed to overcome penetration limitations associated with existing methods. For instance, employing long-wavelength light sources (e.g., near-infrared II) can modestly extend the penetration depth of PDT (approximately 2 cm)[33,34]. Alternatively, the use of high-energy light (e.g., X-rays) to excite scintillator photosensitive materials facilitates deeper penetration[35], although prolonged exposure to high-energy radiation entails potential safety concerns. Direct excitation strategies, such as those using long-afterglow luminescent materials[6], micromagnetic induction LED lights[36], inserted optical fibres[5], chemiluminescence[37] and bioluminescence[18], can effectively mitigate the risks associated with prolonged exposure at a high penetration depth. However, these approaches typically require external energy sources to support or continuously replenish photosensitive reactants. The development of a persistent, long-term light source is critical for maintaining ROS generation at tumour sites. In our study, we utilised the bioluminescence from bacteria as an implantable light source within tumours to continuously activate the photosensitizer NR, thereby enhancing cancer mPDT. Owing to the persistent light emission from PB@MCs after intratumoural implantation, harnessing the chemical energy of host organisms can provide adequate ROS for ongoing PDT activation. This method offers a significant advantage over those requiring external light sources, which typically have limited penetration capability in physiological environments. Although the bioluminescence intensity was considerably lower than that of standard clinical PDT lights or lasers, PB@MC-enhanced Sd-PDT showed superior efficacy, particularly in treating large tumours exhibiting high light absorption, for which traditional PDT light penetration is inadequate and may cause photodamage. Additionally, our findings indicate that treating tumours with a high-dose pulsed light (0.33 W/cm$^2$, the maximum permissible exposure of skin according to

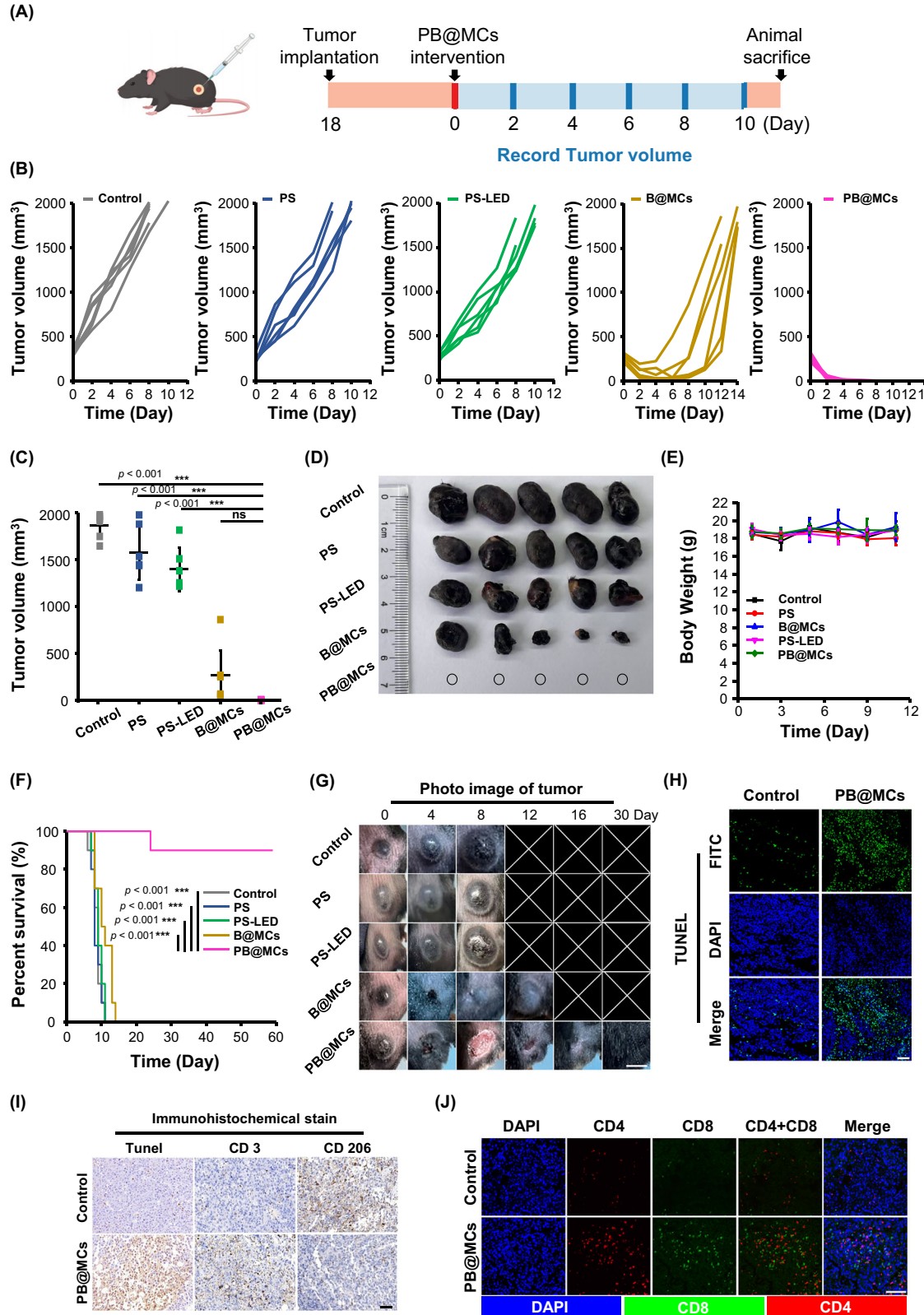

the American National Standard) curbs tumour growth less effectively than treating tumours with PB@MCs, thus supporting the utility of PB@MCs for Sd-PDT.

PB@MCs, a generation microbial ball complex, holds significant promise for clinical applications in cancer treatment. Using ROS detection assays, metabolomics, and immunohistochemistry, we elucidated the mechanisms underlying their potent antitumour effects

(Fig. 8). Initially, tumour cells exploit metabolic reprogramming to achieve uncontrolled proliferation. To maintain the redox balance within the tumour microenvironment, the production of antioxidants, such as glutathione (GSH), is crucial[38]. The existing literature has explored the feasibility of enhancing antitumour effects by exacerbating redox imbalances in tumour cells[39,40]. Our combined results from metabolomics and ROS detection experiments demonstrated

**Fig. 6 | In vivo therapeutic effects of PB@MCs on tumour growth in melanoma mice. A** Schematic representation of PB@MC treatment in melanoma mice. Images are created with biogdp.com. On day 18 post-B16 cell ($1 \times 10^6$/mL) implantation (-300 mm³ in size), 30 μL of PB@MCs ($3.6 \times 10^4$/mL) were intratumourally injected into tumour-bearing mice. **B** Tumour growth curves for treatment groups in vivo. Following treatment, tumour volumes were measured every two days and growth curves were obtained ($n = 6$ mice). **C** Average tumour volumes on day 8 post-treatment ($n = 6$ mice). The data are presented as mean ± SD. Statistical significance is noted with ***$p < 0.001$, compared to the data for PS-LED, PS, B@MCs and control group according to one-way ANOVA with Tukey's multiple comparisons test performed using GraphPad Prism 9 XML project software. ns not significant. **D** Optical photographs of the dissected tumours according to the treatment type. Mice were sacrificed on day 8 post-treatment for tumour visualisation. Ex vivo images were captured using a Canon ZHS2402 camera (Japan).

**E** Changes in the body weight of mice during in vivo treatment for 11 days ($n = 6$ mice). The data are presented as mean ± SD. **F** Kaplan–Meier curves showing the survival rates of the indicated mice at 60 days ($n = 10$ mice). The data are presented as ***$p < 0.001$, compared to the data for other treatment groups of tumour-bearing mice according to the log rank performed using GraphPad Prism 9 XML project software. **G** Optical photographs of tumour-bearing mice subjected to different treatments for 30 days. Scale bar: 10 mm. **H** TUNEL staining of tumour tissues across treatment groups in which green fluorescence represents FITC-labelled deoxyuridine triphosphate (dUTP) detected in apoptotic cells. Scale bar: 200 μm. **I** Representative immunohistochemical staining images (TUNEL, CD3, CD206, and CD86) of various treatment groups after 1 day of treatment. Scale bar: 100 μm. **J** CD4⁺ and CD8⁺ immunofluorescence staining of tumour tissues in various treatment groups.Scale bar: 200 μm. Source data are provided as a Source data file.

that PB@MCs significantly activated ROS and inhibited GSH, ultimately inducing oxidative stress in tumour cells. The mechanisms by which PB@MCs lead to GSH depletion were further investigated. Metabolomic analyses revealed that the GSH metabolic pathway was suppressed after treatment with PB@MCs, along with the downregulation of amino acid metabolism and the pentose–phosphate pathway, which reduced the amount of substrate necessary for GSH synthesis and regeneration. Additionally, pathways related to glutamine metabolism, TCA cycle, and oxidative phosphorylation were inhibited, indicating that the primary energy supply routes for tumour cells were disrupted[41]. Furthermore, the suppression of purine, pyrimidine, and lipid metabolism suggests that tumour cells lose the ability to synthesise the essential membrane structures and genetic material necessary for proliferation. Moreover, an increase in IFN-γ expression; the increase in the ratio of CD8⁺ T cells[42,43] and mature DCs[44]; and the reduced M2-like macrophage infiltration in tumour sections[45] collectively indicate that PB@MCs activate the antitumour immune microenvironment within tumour tissues. While V.H.BB170 treatment alone (B@MCs) showed only partial efficacy in tumour suppression, its combination with the photosensitizer NR-boosted mPDT was markedly effective. This strategy of integrating living bioluminescent bacteria with photosensitizer-enhanced mPDT simultaneously inhibited tumour growth and prevented tumour recurrence by activating both innate and adaptive immune responses. Our findings highlight the utility of strategically integrating bioluminescent bacteria with mPDT with immunotherapy, leveraging bioluminescent bacteria-activated PS as a durable source of ROS for sustained mPDT activation and an influential immunostimulatory agent that amplifies PDT-induced ICD. We are optimistic that this approach, combining bioluminescent bacteria and enhanced mPDT with the immunostimulatory capabilities of bacteria, offers significant potential for clinical applications owing to its exceptional anti-tumour efficacy and acceptable safety profile.

In summary, we developed an Sd-PDT system which enables continuous and uniform light emission throughout a tumour without requiring an external energy source or continuous replenishment of photosensitive reactants. By utilising nutrients within the tumour microenvironment, this system produces prolonged, low-dose light emission that can be sustained for up to 50 h, which significantly surpasses the duration of all currently available chemically driven PDT methods. Moreover, PB@MCs significantly prolonged the self-supported generation of ROS. A single injection of PB@MC is required to effectively eradicate large tumours, including melanoma and hepatocarcinoma tumours in mice and rabbits. Furthermore, our findings indicate that PB@MC-enhanced Sd-PDT not only suppresses tumour growth but also evokes strong antitumour immunity. Consequently, the successful development of Sd-PDT offers significant potential for advancing tumour biology studies and cancer treatments. This method may emerge as a promising generalised therapeutic strategy for treating various cancer types and tumour sizes.

## Methods

### Ethics statement
All animal experiments were performed in accordance with the guidelines approved by the Laboratory Animal Welfare and Ethics Committee of The Second Affiliated Hospital of Zhejiang University School of Medicine (ARIB-2023-1520). The maximum tumor size permitted by the ethics committee/institutional review board is 2 cm³ for mice or 60 cm³ for rabbit. The tumor sizes in all experiments have never exceeded this threshold.

### Reagents and materials
Sodium alginate (a molecular weight of 460 kDa; molar ratio of mannuronic acid to guluronic acid ratio, 2:1) was procured from Qingdao Bright Moon Seaweed Group Co., Ltd. in Qingdao, China. Chemical reagents such as dimethyl sulfoxide (DMSO), methanol, and ether were sourced from Sinopharm Chemical Reagent Co., Ltd. Anhydrous calcium chloride ($CaCl_2$; Thermo Fisher Scientific, C614-500); dibasic sodium phosphate ($Na_2HPO_4$; ≥98.5%, Cat. 7558-79-4) were obtained from Calbiochem®. The Annexin V-FITC/PI apoptotic cell death assay kit (abs50001) was acquired from Absin Bioscience Inc., Shanghai, China. Corning, New York, USA, supplied the PC membrane Transwell-24 (Cat. 3422) and Transwell-6 (Cat. 3452). The 2216E medium (catalog No. bio-54545) was purchased from the Institute of Hydrobiology, Wuhan, China, and CCK-8 assay kits came from Target Molecule Corp., USA. LIVE/DEAD staining kits, penicillin, streptomycin, trypsin-EDTA, and DMEM were sourced from ThermoFisher Scientific, Grand Island, NY, and Hyclone Laboratory, South Logan, Utah, USA, respectively. RPMI 1640 medium and fetal bovine serum (FBS) were procured from Gibco and Gemini, Woodland, USA, respectively. Beyotime Biotechnology, Shanghai, China, provided the ROS Assay Kit and Hoechst 33258. Dibenzocyclooctyne-PEG4-N-hydroxysuccinimidyl (DBCO-PEG4-NHS) ester was acquired from Xi'an Ruixi Biological Technology Co., Ltd. The cellular reactive oxygen species detection assay kit (DCFH-DA) was ordered from Abcam. Propidium iodide (PI) and 4',6-diamidino-2-phenylindole (DAPI) were obtained from Shanghai Yuanye Beyotime Co., Ltd, China. Pure water (resistivity: 18.2 mΩ·cm) was produced using an ELGA Purelab classic UVF system for preparing working solutions and buffers, while sodium hydroxide (NaOH, ≥ 98.0%) was purchased from Sigma-Aldrich. Abcam supplied the Alexa Fluor® 488-conjugated CRT primary antibody (ab196158, British). Poly-L-Lysine (MW: 15,000-30,000, P4832), BMS-345541 (Cat. 401480), were purchased from Sigma-Aldrich (St. Louis, MO, USA). Chemstan (CS-13629, China) provided Etoposide VP-16. Tissue •OH Assay Kit (BBox-Probei O28, BB-46072, BestBio, Shanghai, China); tissue freezing medium (14020108926, Leica, Germany); the ELISA kit for IFN-γ measurement (Solarbio, SEKM-0145, China); anti-CD4-PE (Biolegend, Cat. 100511), anti-CD45-APC-Cy7 (Cat.557659), anti-CD3-PerCP-Cy5.5 (Cat. 551163), anti-CD11c-FITC (Cat. 117305), anti-CD80-PE (Cat. 560016), and anti-CD8-FITC (Cat. 553030) were purchased from BD, USA. Details of all antibodies used are provided in Supplementary Table 1

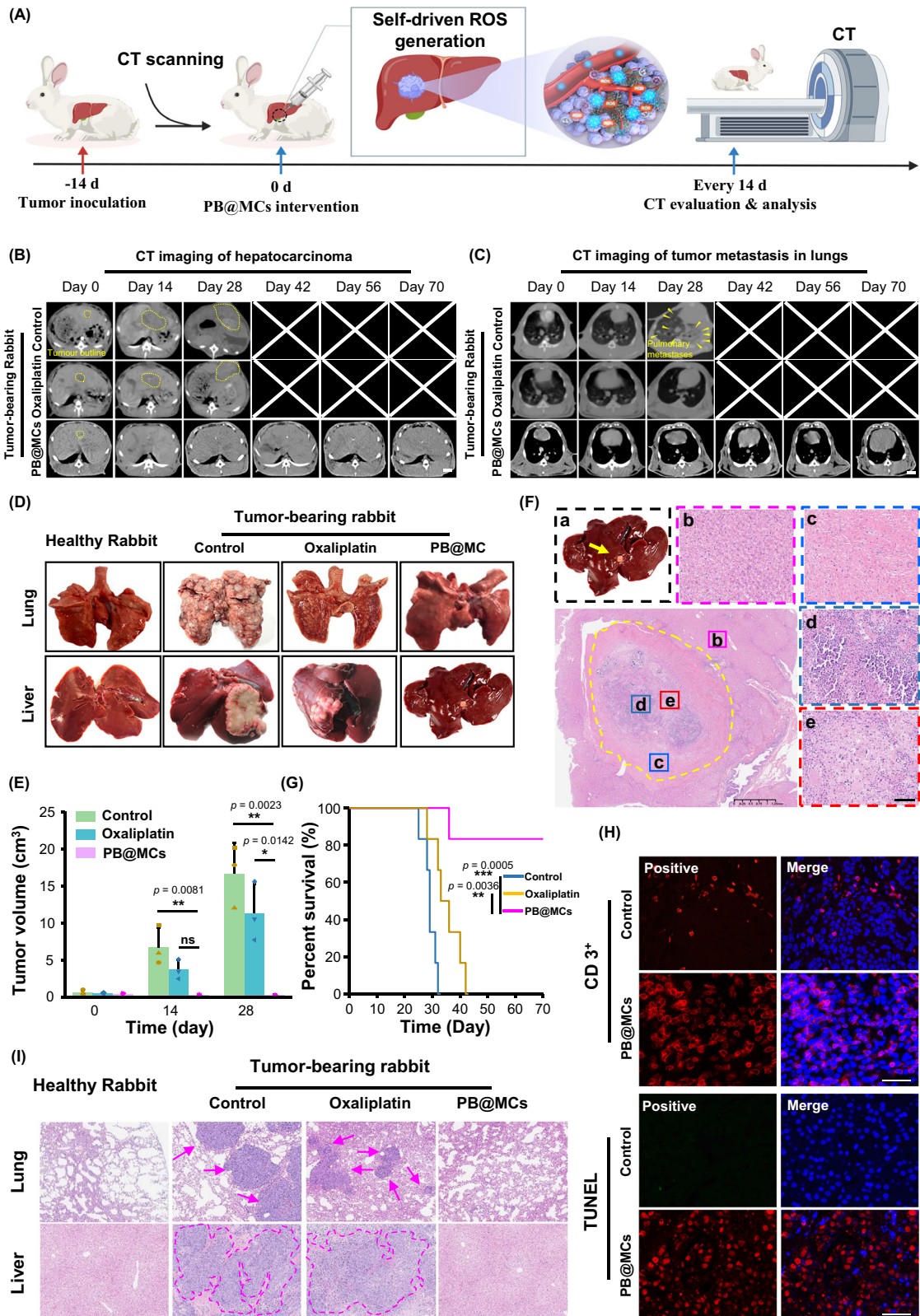

## Cultural conditions of bacteria

*Vibrio harveyi BB170* (catalog No. bio-108879), *Aliivibrio fischeri-bio115653* (catalog No. bio-105653), and *Aliivibrio fischeri-7744* (catalog No. bio-72794) were obtained from Biobw, Beijing, China, and acclimatized under controlled physiological conditions by varying growth factors, including temperature (25–37 °C) and medium composition (2216E to DMEM). These modifications ensured the survival of

the bioluminescent bacteria under physiological conditions. A sterile pipette was employed to transfer approximately 0.5 mL of 2216E medium into a 2 mL Eppendorf tube (Centrifuge 5810 R, Eppendorf). This reconstituted bacterial suspension was incubated overnight at 37 °C in a sterile, clear BeyoGold™ Bacteria Culture Tube containing 5 mL of 2216E medium, agitated at 10 × g. Simultaneously, 100 μL of the suspension was cultured on a 2216E agar plate at 37 °C for 12–16 h

**Fig. 7 | In vivo therapeutic effects of PB@MCs on tumour growth and metastasis in rabbit hepatocarcinoma. A** Schematic of PB@MC treatment in hepatocarcinoma-bearing rabbits. Images are created with biogdp.com. **B** CT images depicting tumour growth in vivo. Tumours in rabbits treated with and without PB@MCs were compared with oxaliplatin-treated groups using CT imaging at predefined intervals (0–70 days). Tumour margins are highlighted with a yellow outline. Scale bars: 1 cm. **C** CT images of pulmonary tumour metastases. Lungs were imaged at the same intervals post-treatment to identify pulmonary metastases, indicated by yellow arrows. Scale bars: 1 cm. **D** Ex vivo photographs of livers and lungs from healthy, tumour-bearing control, PB@MCs, and oxaliplatin-treated rabbits. Animals were sacrificed on day 28 post-treatment. **E** Tumour dimensions were quantified over a 28-day post-treatment ($n = 3$ rabbits). The data are presented as mean ± SD. Statistical significance is noted with **$p < 0.01$, *$p < 0.05$ compared to the data for oxaliplatin-treated and control group according to one-way ANOVA test. **F** H&E staining of liver tumour sites treated with PB@MCs. The annotations include purple squares (b) indicating normal liver cells, blue squares (c) indicating fibrotic tissue, cyan squares (d) indicating necrotic liver cells, and red squares (e) indicating swollen liver cells. Scale bar: 100 μm. **G** Kaplan–Meier curves showing the survival rates of the indicated mice at 70 days ($n = 6$ rabbits). The data are presented as ***$p = 0.0005$ compared to the data for control groups and **$p = 0.0036$ compared to the data for oxaliplatin-treated groups according to the log rank test performed using GraphPad Prism 9 XML project software. **H** CD3⁺ immunofluorescence (red fluorescence represents Cy5.5-labeled anti-CD3-PerCP and blue fluorescence DAPI represents the nucleus) and TUNEL staining red fluorescence represents Cy5.5-labeled dUTP and blue fluorescence DAPI represents the nucleus. Scale bar: 50 μm. **I** H&E staining of hepatocarcinoma tumours and pulmonary metastases. Purple arrow: pulmonary metastasis, Scale bar: 250 μm. Source data are provided as a Source data file.

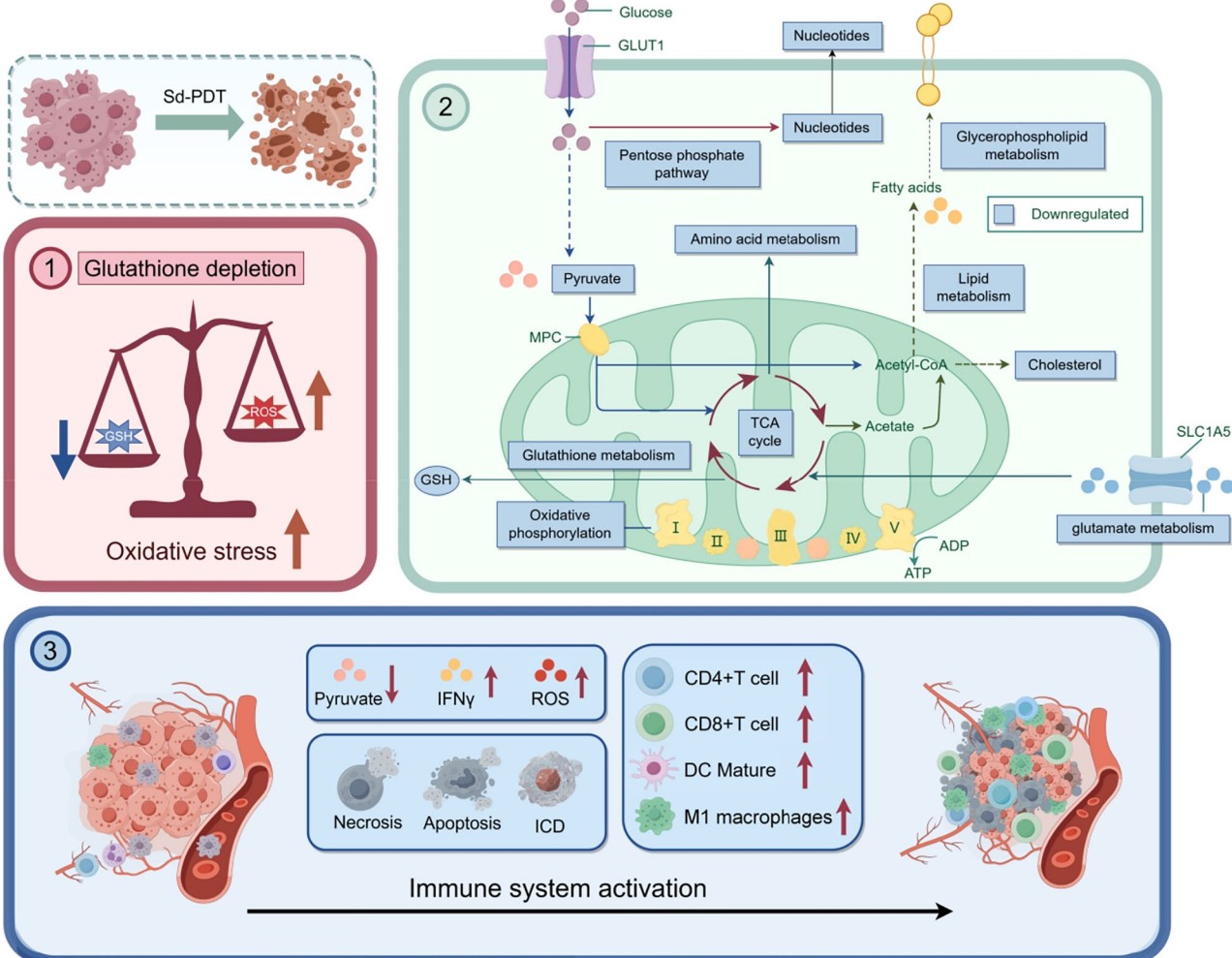

**Fig. 8 | Mechanisms of anti-tumour effects of PB@MCs.** PB@MCs inhibit the malignant biological behaviour of tumours through several mechanisms. (1) PB@MCs disrupt the redox balance in tumour cells. This process is triggered synchronously by the production of reactive oxygen species (ROS) output and glutathione depletion. (2) PB@MCs lead to metabolic reprogramming of tumour cells. The synthesis pathways of biomolecules essential for tumour cell proliferation, such as phospholipids and nucleic acids, were significantly suppressed. Furthermore, energy metabolism pathways related to tumour cells, including glycolysis, oxidative phosphorylation, and glutamine metabolism, were markedly inhibited. This results in a significant reduction in the synthesis of glutathione, a key metabolite that alleviates oxidative stress, making it difficult for tumour cells to maintain their malignant phenotypes associated with unlimited proliferation. (3) PB@MCs facilitate metabolic-immune co-evolution of the tumour microenvironment (TME), transforming it from an immunosuppressive to an immunoactive state. The content of lactic acid, a marker of immune-suppressive TME, significantly decreased, while levels of INFγ and ROS, markers of immune activation, rose significantly. This process relieved the exhaustion of CD4⁺ T cells, CD8⁺ T cells, and mature dendritic cells (DCs) in the tumour immune microenvironment (TIME), reactivating their anti-tumour functions and promoting the conversion of tumour-associated macrophages (TAM) from the M2 to the M1 type. Ultimately, this cascade of events induced tumour cell necrosis, apoptosis, or immunogenic cell death. Images are created by figdraw.com.

to cultivate monoclonal colonies. The optical density (OD) at 600 nm was periodically measured to establish the bacterial growth curve. Once the bacteria reached the logarithmic phase, the strain was inoculated into a fresh culture medium. For CFU/mL calculation, 100 μL of the bacterial suspension was taken when the bacteria reached the logarithmic phase (OD value of around 0.8) and dilute stepwise to 16-fold. Dilutions (0, 2, 4, 8 and 16) of the bacterial suspension were added to a 96-well plate to recorded the OD600 values. Subsequently, 20 μL of five different proportions of bacterial suspension was mixed with 180 μL PBS buffer and further diluted stepwise from $10^{-1}$ to $10^{-8}$. Then, 100 μL of bacterial suspension was cultured on a 2216E agar plate. After overnight incubation, monoclonal bacterial colonies were counted. Finally, a standard linear graph of bacterial count (CFU/mL) was plotted against the OD600 reading.

## PB@MC construction

*Vibrio harveyi BB170* was collected by centrifugation at $100 \times g$ for 10 min using a Centrifuge 5810R (Eppendorf) in a 15 mL Eppendorf tube and subsequently resuspended in 2216E medium for PB@MC construction. The bacterial suspension was merged with a 1.5 wt% sodium alginate solution. After agitating the mixture, it was extruded through a 5 mL syringe equipped with a 0.5 mm needle into a 0.2 M CaCl$_2$ solution (gelling bath) using an electrostatic droplet generator to form alginate-Ca beads. These beads were then soaked in a 0.05 wt% poly-L-lysine solution (volume ratio 1:1) for 10 min to form a poly-L-lysine coating. The NHS-Neutral Red photosensitizer (NHS-NR) was produced through Cu-free click chemistry, 200 μL of the NR-N$_3$ stock solution (25 mM) was added to 2 mL of DBCO-PEG4-NHS ester (50 mM), which was freshly prepared by dissolving it with dry DMSO. The reaction was incubated at RT for 4 h and then dialyzed against PBS for purification. For conjugation, 500 μL of poly-L-lysine-coated bacteria@MCs (PLL-B@MCs) at $3.6 \times 10^4$/mL was mixed with 2 mg/mL NHS-NR (500 μL, 2.2 mM) at 37 °C for 20 min. Subsequently, the medium was renewed with 2216E medium. The optical density was evaluated at 452 nm to quantify the unreacted NHS-NR (extinction coefficient $\varepsilon_{max} = 27,500$ M$^{-1}$·cm$^{-1}$), revealing a binding efficiency of approximately 49%. The encapsulated microcapsules were reinstated in 15 mL of 2216E medium and analyzed under a microscope. All formulated PB@MCs were cultured at 35–37 °C for subsequent applications in a HerryTech KE-200 incubator, Shanghai, China.

## Characterization of PB@MCs

The morphology and size of empty MCs, bacteria@MCs (B@MCs), poly-L-lysine coating bacteria@MCs (PLL-B@MCs), and PB@MCs were determined by microscopy (FV1200, Japan). For self-bioluminscence testing, empty MCs, B@MCs and PB@MCs at (500 μL, $3.6 \times 10^4$/mL) were added to 24-well chambers to assess their self-bioluminescence activities using CLSM (LSM900 Carl Zeiss, Germany). Images were collected with 5× and 10× objectives at the 488–620 nm channel.

For long-term bioluminscence (BL) intensity testing of PB@MCs, the tested PB@MCs (50 μL, $3.6 \times 10^4$/mL) were incubated in 96-well black plates (FCP966, Beyotime Biotechnology, China) with 150 μL 2216E medium. The BL intensity was continuously monitored at 37 °C every hour, with light emission detected by a Microplate Reader (Tecan Spark, Switzerland) until the BL reached its maximum intensity. After removal of the residual 2216E medium, DMEM medium (150 μL) and B16 tumour homogenate (150 μL, diluted 1:1 with PBS) were added when the BL intensity decreased. BL intensity was recorded for another 50 h.

For photo imaging of PB@MCs, 6 mL of freshly prepared PB@MCs were suspended in 50 mL Eppendorf tubes (Centrifuge 5810 R, Eppendorf) with 20 mL of 2216E medium at 35 °C, in an incubator (HerryTech KE-200, Shanghai, China) until self-bioluminescence reached its maximum intensity. After removing the residual 2216E medium, 2 mL of PB@MCs were suspended in a 14 mL bacteria culture

tube (BeyoGold™ Bacteria Culture Tube, Clear, Sterile) and refreshed with PBS (3 mL), DMEM medium (3 mL), and B16 tumour homogenate (3 mL, diluted 1:1 with PBS) when the BL intensity decreased. BL intensity was recorded for another 32 h using a Xiaomi 14 smartphoto with an exposure time of 5 s.

For self-BL spectra of B@MCs and PB@MCs, 500 μL, $3.6 \times 10^4$/mL B@MCs and PB@MCs were suspended in 3.5 mL quartz glass cuvette (s2840-04-2EA, aladdin, China) with equal volumes of DMEM to analyze their self-BL emission spectrum from 300 to 700 nm without excitation using an Edingbour NanoSpectralyzer fluorimetric analyzer (Applied NanoFluorescence, FLS980). The data was fit to a Gaussian function using OriginLab 8.0 software.

To obtain the in vitro ROS release profile, freshly made PB@MCs (30 μL, $3.6 \times 10^4$/mL) were incubated in 96-well black plates with 150 μL of 2216E medium. The BL intensity was continuously monitored at 37 °C every hour, with light emission detected by a Microplate Reader (bioluminescence model) until the BL reached its maximum intensity. After removal of the residual 2216E medium, DMEM medium (100 μL) was added to each well of the 96-well black plates. For comparison, the photosensitizer NR (0.32 mM) in 100 μL of DMEM with LED irradiation (450 nm laser LED, Xi'an Lei Ze Electronics Tech Co., Ltd, Shanxi, China) was used as a tradional PDT control group[46] (LED radiation). The LED radiation group in 96-well plates were exposed to 300 mW/cm$^2$ LED radiation for four intervals (20 min each, 15 min On and 5 min Off). DCFH was pre-prepared by dissolving DCFH-DA (1 mM, 1 mL) in NaOH (0.01 M, 4 mL) at room temperature for 30 min, then balanced with Na$_2$HPO$_4$ (25 mM, 20 mL) to make a stock solution at 40 μM in the dark. ROS generation was determined by adding 40 μL of DCFH (40 μM) at 0, 0.25, 0.5, 0.75, 1, 2, 5, 12, 16, 18, 24, 28, 30, 36, 40 and 48 h. After 15 min of incubation, the residual medium was transferred to a 96-well black plate, and fluorescence intensity of DCF was immediately recorded by a Microplate Reader (Tecan spark, Switzerland) at Ex/Em: 488/525 nm. Fresh DMEM and 40 μL DCF were also included as negative and positive control.

For In vitro PB@MCs energy radiation density, PB@MCs (100 μL, $3.6 \times 10^4$/mL) were incubated in 24-well black plates (3 wells in parallel) with 500 μL of 2216E medium. The BL intensity was continuously monitored at 37 °C every 2 h, with light emission detected by a Microplate Reader (Tecan Spark, Switzerland) until the BL intensity reached 100, 200, 500, 700 and $1000 \times 10^4$ cps. The energy radiation density of PB@MCs was immediately recorded using a Laser Power Meter (YanbangTech VLP2000, Beijing, China) with a distance of -0.2 cm between the detector and the plate surface. The plate was wrapped in aluminum foil with a hole in the center of each well for light transmission detected by the Laser Power Meter. All testing was conducted in the dark.

## Cell culture

A375 (Cat. CL-0014), Hep3B (Cat. CL-0102), and B16 (Cat. CL-0319) cell lines were procured from Procell system (Wuhan, China). VX2 (Cat. BFN60700420) cell line were procured from BLUEFBIO™ Product (Shanghai, China). and cultured in RPMI 1640 medium or DMEM, supplemented with 10% FBS (Gemini, Woodland, USA) and 1% penicillin-streptomycin solution (Hyclone Laboratory, 10,000 U/mL) in a cell culture incubator. A375, Hep3B, B16 and VX2 cells were morphologically confirmed according to the information provided by the cell-source center. STR analysis was performed to authenticate A375, Hep3B, B16 cells in Supplementary Tables 2–4. STR loci are amplified using fluorescently labeled PCR primers that flank the hypervariable regions. For VX2 cells, a certificate of analysis were performed in Supplementary files. The cells were all negative in mycoplasma test. Bone Marrow (BM)-Derived Dendritic Cells (BMDCs) were produced from the BM of 8-week-old C57B6 mice. The procedure involved cutting the hind legs of the mouse, removing the attached tissue, and sterilizing the femur and tibia by soaking them in 75% alcohol for

5–10 s. The bones were then washed with PBS, and both ends were cut off. A syringe needle was inserted into the bone to flush the BM into the medium. The BM extract was carefully agitated, and cells were separated by centrifugation at $400 \times g$ for 5 min. The cell pellet was dissolved, and the cell density regulated to $1 \times 10^6$/ml prior to being introduced into RPMI 1640 medium containing recombinant mouse GM-CSF (20 ng/ml) and IL-4 (10 ng/mL). The medium was exchanged every 72 h, and immature DCs were harvested on day 8.

## Cell viability test

Tested cells were plated in 24-well transwell plates at $5 \times 10^4$ cells per well. Following an overnight incubation, the culture medium was substituted with 1 mL of fresh medium, to which 200 μL of PBS, empty MCs connected to a photosensitizer (P@MCs, contain 200 μg, 0.22 μmol NR), MCs containing V.H.BB170 bacteria (B@MCs, $1 \times 10^3$ bacteria cells per MC), and PB@MCs ($1 \times 10^3$ bacteria cells per MC and contain 200 μg, 0.22 μmol NR) were introduced into the upper chamber. The cells were then incubated at 37 °C for eight hours. Subsequent to the removal of the supernatants and any residual P@MCs, B@MCs, and PB@MCs, each well received 500 μL of a 10% CCK-8 solution (diluted in DMEM). The plates were further incubated at 37 °C for an additional hour. Absorbance was recorded at OD 450 nm using a Microplate Reader (Tecan Spark, Switzerland). Cell viability was calculated (1):

$$\text{Cell viability\%} = \frac{(A_N - A_B)}{(A_C - A_B)} \times 100\% \qquad (1)$$

where $A_N$, $A_C$ and $A_B$ represent the absorbance at 450 nm of the treated, untreated, and blank samples, respectively.

## Confocal microscopy imaging

For LIVE/DEAD staining. B16, VX2, Hep3B, and A375 cells were cultured in 24-well transwell plates at $5 \times 10^4$ cells per well and incubated for 24 h. Subsequently, the supernatants were exchanged with fresh RPMI 1640 or DMEM medium, and the cells were subjected to 200 μL of P@MCs(P@MCs, (contain 200 μg, 0.22 μmol NR), B@MCs ($1 \times 10^3$ bacteria cells per MC), and PB@MCs ($1 \times 10^3$ bacteria cells per MC and contain 200 μg, 0.22 μmol NR) in the upper chamber for 8 h. After treatment, the cells were twice washed with PBS and stained using a Calcein/PI viability assay kit, diluted in culture media, for 20 min at 37 °C. All cell samples were examined under a confocal laser scanning microscope (FV 1200, Olympus, Japan) utilizing 5× or 10× immersion objective lenses at excitation wavelengths of 517 nm and 617 nm.

In vitro ROS content detection. To assess intracellular ROS generation, a DCFH-DA staining kit was employed. Hep3B and A375 cells were cultured in 24-well transwell plates at 37 °C with 5% $CO_2$ at $5 \times 10^4$ cells per well after an overnight pre-incubation. Fresh RPMI 1640 or DMEM medium containing 10% v/v FBS replaced the supernatants. The cells underwent treatment with a vehicle control (no treatment), PB@MCs (100 μL, $3.6 \times 10^4$/mL, contain 100 μg, 0.11 μmol NR), B@MCs (100 μL, $3.6 \times 10^4$/mL), and P@MCs (100 μL, $3.6 \times 10^4$/mL, contain 100 μg, 0.11 μmol NR) in the upper chamber in darkness at 37 °C for 2 h. Subsequently, the cells were incubated in fresh medium containing DCFH-DA (10 μM) for 30 min, washed thrice with PBS, and stained with 10 μg/mL Hoechst 33342 (Beyotime Biotechnology, Shanghai, China) in PBS for five minutes. DCF fluorescence was promptly visualized using CLSM (LSM900 Carl Zeiss, Jena, Germany) at 488 nm.

In vivo ROS content detection. To evaluate the •OH levels produced by PB@MCs in mouse tumour tissues in vivo. Four groups of B16 melanoma-bearing mice were injected intratumourally with 50 μL PBS (control), B@MCs, P@MCs or PB@MCs. After 10 h,

tumours were aseptically collected, weighed, and processed according to the manufacturer's instructions. Tumour tissues were homogenized by cryo-grinding at 60 Hz for 120 s, then adjusted to 50 mg/mL with sterile PBS. Subsequently, 190 μL of tumour homogenate (50 mg/mL) and 10 μL of BBoxiProbe O28 working solution were added to a 96-well black plate and incubated at 37 °C in the dark for 20 min. Fluorescence intensity was measured using a microplate reader (excitation: 488 nm; emission: 520 nm). Protein concentrations in the homogenate were quantified using the Bradford Protein Assay Kit (P0006, Beyotime Biotechnology, Shanghai, China). Tissue •OH levels were expressed as fluorescence intensity normalized by protein concentration.

## Flow cytometry analysis

To evaluate if PB@MCs treatment prompted apoptotic cell deaths, B16, VX2, Hep3B and A375 cells ($5 \times 10^5$ cells/well) were cultured in 6-transwell plates using RPMI 1640 or DMEM for 16 h. Fresh media containing 400 μg/mL etoposide (serving as the positive control for the VP-16 group) and 400 μL of P@MCs (contain 400 μg, 0.44 μmol NR), B@MCs ($1 \times 10^3$ bacteria cells per MC), or PB@MCs ($1 \times 10^3$ bacteria cells per MC and contain 400 μg, 0.44 μmol NR) were then introduced to the upper chamber for 8 h. Following incubation, cells were harvested and stained using an Annexin V-FITC/DAPI apoptosis kit, then analyzed via flow cytometry.

To measure IFN-γ levels, whole blood was collected from 8-week-old C57B6 mice by eyeball blood collection on day 3 after treatment, including an intratumoural injection of 30 μL saline, intratumoural injection of 1.5 mg/kg PS (NHS-NR, 0.033 μmol, 30 μL) (PS group), PS coupled with 300 mW/cm² blue light LED radiation for 60 min (PS-LED group), intratumoural injection of 30 μL ($3.6 \times 10^4$/mL) of bacterial encapsulate of MCs (B@MCs), and PS-modified B@MCs (PB@MCs group). Blood was centrifuged at $100 \times g$ for 5 min, and the serum was extracted. Levels of IFN-γ were quantified using an ELISA kit (Solarbio, SEKM-0145, China) as per the provided instructions.

For CRT expression assay. CRT expression was assessed through flow cytometry. A375 and B16 cells were plated in 6-well plates at $5 \times 10^5$ cells per well and incubated for 24 h. Treatments administered included PS (NHS-NR, 0.033 μmol, 30 μL/well) (PS group), PS in conjunction with 300 mW/cm² of blue light LED radiation for 60 min (PS-LED group), and intratumoural injections of 30 μL of bacterial encapsulate of MCs (B@MCs), and 30 μL of PS-modified B@MCs (PB@MCs group) for 6 h. Post-treatment, cells were rinsed with cold PBS (1 mL) at 4 °C and fixed in 0.25% paraformaldehyde (1 mL) for 10 min, then incubated with Alexa Fluor®488-conjugated CRT primary antibody (ab196158, Abcam, UK) for 30 min before undergoing flow cytometry.

To explore in vitro DC maturation, BMDCs were cultured from the BM of 8-week-old C57B6 mice. B16 cells were pre-treated with the PS group (NHS-NR, 0.033 μmol, 30 μL), PS-LED group, 30 μL of B@MCs, or 30 μL of PB@MCs at the same concentration for 6 h. Subsequently, $1 \times 10^6$ immature DC cells were co-cultured with these pre-treated B16 cells for 24 h. The maturation of DC cells was then assessed by staining with anti-CD11c-FITC, anti-CD80-PE, and anti-CD86-APC antibodies and analyzed using flow cytometry (BD, FACSCanto II). Flow cytometry gating strategy for the analysis of DC were shown in Supplementary Fig. 22.

For intratumoural infiltration of T lymphocytes. To prepare a tumour single-cell suspension, tumours were excised on day 3 post-treatment, sectioned into small pieces, and enzymatically digested in DMEM containing 1 mg/mL collagenase IV (C8160, Solarbio, China) and 0.2 mg/mL DNase I (D8071, Solarbio, China) for 45 min at 37 °C. Isolated cells were stained with anti-CD45-APC-Cy7, anti-CD3-PerCP-Cy5.5, anti-CD4-PE, and anti-CD8-FITC antibodies for 30 min and subsequently analyzed by flow cytometry. Flow cytometry gating strategy for the analysis of T cells were shown in Supplementary Fig. 22.

## Metabolomics analysis

Mice inoculated with B16 cells were prepared for further interventions once tumours reached 300 mm³. The qualified subjects were randomly allocated into five groups ($n = 5$ per group) to receive treatments including an intratumoural injection of 30 μL saline, intratumoural injection of 1.5 mg/kg PS (NHS-NR, 0.033 μmol, 30 μL) (PS group), PS coupled with 300 mW/cm² blue light LED radiation for 60 min (PS-LED group), intratumoural injection of 30 μL ($3.6 \times 10^4$/mL) of bacterial encapsulate of MCs (B@MCs), and PS-modified B@MCs (PB@MCs group). Each animal was administered a single intratumoural injection, anesthetized with a lethal dose of sodium pentobarbital (400 mg/kg), and euthanized 24 h later. tumour samples were then stored in liquid nitrogen for subsequent metabolomics analysis.

## Animal treatment

Healthy female C57B6 mice (6–8 weeks old, 20 g) were acquired from the Experimental Animal Center at Hangzhou Medical College, China. Female New Zealand White rabbits (6 months old, weighing between 2 and 2.5 kg) were procured from Qingdao Kangda Rabbit Co., Ltd. (Qingdao, China). All animals were maintained under controlled environmental conditions (22–25 °C), with a 12-h light/dark cycle and relative humidity ranging from 40 to 70%, in plastic cages using sterilized wood shavings as bedding. The B16 melanoma tumour model was established in mice by subcutaneously injecting 100 μL of B16 cell suspension ($1 \times 10^6$ cells) into the hind leg. Tumour volumes were recorded bi-daily using a specific formula:

$$\text{Tumor volume(mm}^3) = width^2 \times \frac{length}{2} \qquad (2)$$

Once tumours reached roughly 300 mm³, preparations for further interventions began. Qualified animals were randomly distributed into five groups for various treatments, which included a single intratumoural injection of either 30 μL saline (vehicle control), 1.5 mg/kg PS (NHS-NR, 0.033 μmol, 30 μL) (PS group), PS combined with 300 mW/cm² of blue light LED radiation for 60 min (PS-LED group), 30 μL ($3.6 \times 10^4$/mL) of bacterial encapsulate in MCs (B@MCs), or PS-modified B@MCs (PB@MCs group), all administered on day 18.

For inducing hepatic VX2 tumours in rabbits, VX2 cell suspensions ($2 \times 10^6$ cells, 200 μL) were implanted in the thigh muscles of donor rabbits. When tumours grew beyond 2 cm, typically within two weeks, donor rabbits were euthanized via intravenous administration of a lethal dose (2 mL/kg) of xylazine hydrochloride for tumour harvesting. Each tumour was meticulously minced into 1 mm³ pieces under sterile conditions. Recipient rabbits were sedated using intramuscular injections of xylazine hydrochloride (250 μL/kg), and the minced tissue was precisely inserted into the subcapsular parenchyma of the left hepatic lobe guided by a 16-slice CT spiral scan (Brilliance-16, Philips, USA). tumour development was monitored by CT until they reached about 1 cm³. Hepatocarcinoma-bearing rabbits of similar tumour sizes were randomly assigned to three groups: vehicle control, oxaliplatin treatment, and a PB@MCs group. Each received a singular intratumoural injection of oxaliplatin (100 μL, 6.62 mg/rabbit) or PB@MCs suspension (500 μL, $3.6 \times 10^4$/mL) on day 14. Tumour growth and lung metastasis were periodically assessed via CT (MHCT Brilliance 16, Philips, Holland). Harvested organs were documented using a Canon camera (Japan). Both species were ultimately euthanized using an overdose of sodium pentobarbital (400 mg/kg) to collect tumours and organs for cytokine quantification or for preservation in liquid nitrogen. Rabbit tissues were processed for H&E staining and immunostaining for TUNEL and CD3 markers, whereas mouse tissues were subjected to immunostaining for TUENL, CD4, CD8, CD3, and CD206 markers.

## In vivo biosafety assessment

Healthy female C57B6 mice (age: 6–8 weeks; weight: $20 \times g$) were randomized into four groups. Mice underwent intraperitoneal or subcutaneous injections of either PB@MCs suspensions (30 μL, $3.6 \times 10^4$/mL) or an equivalent volume of PBS. Post-injection observations of behavior and physical condition were conducted periodically. Mice were euthanized via $CO_2$ inhalation on days 1, 3 and 7 to procure organs and blood. Collected tissues were subsequently prepared for H&E staining.

## Statistics and reproducibility

All experiments were repeated at least thrice with three to ten replicates. Data were expressed as mean ± standard deviation (SD) from at least three replicates. All confocal imaging and immunohistochemical staining imaging were repeated at least three replicates. Data analysis was performed by two-tailed Student's $t$-test, one-way ANOVA or Log rank test via by GraphPad Prism 9 XML project software. The difference was regarded as statistical significance if $p < 0.05$.

## Reporting summary

Further information on research design is available in the Nature Portfolio Reporting Summary linked to this article.

## Data availability

The authors declare that all data needed to support the findings of this study are provided within the article, Supplementary information, and Source data file. The metabolism analysis data reported in this paper have been deposited on a public repository, Zenodo Dataverse, by following https://doi.org/10.5281/zenodo.16792981. Source data are provided with this paper.

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

## Acknowledgements

The research was supported by the National Natural Science Foundation of China (No. 22161132027, No. 82302375, No. U23A202181, No. 82272860, and NO. 82472844), the Key Research and Development Program of Zhejiang Province (Program No. 2024C03143), Central Guidance on Local Science and Technology Development Fund of Zhejiang Province (2024ZY01006, 2025C02134), Key Project of Traditional Chinese Medicine Science and Technology Plan of Zhejiang Province(GZY-ZJ-KJ-24077), Special Financial Support for Zhejiang Traditional Chinese Medicine Innovation Teams.

## Author contributions

Z.M. W.W. Y.D. and W.L.W. conceived the idea and designed the experiments. W.W. performed most experiments. W.W. Q.H. and J.Z. constructed the PMCs. W.W. performed cell proliferation and cell imaging. Z.L. and M.H. performed rabbit tumour implantation experiments. Z.L. and J.J. performed mice tumour implantation experiments. B.Y., Y.L. and W.W. contributed in T and DC cell exatraction as well as activity test. B.Y., Y.L. and W.W. tested ROS level in vivo and in vitro. B.Y. and Y.L. quantified the CRT and IFN-γ level in cells and tumours. W.W. and Y.L. contributed in cell viability test. Z.M., Y.D. and W.W. contributed the idea of immunotherapy in the hyperoxia microenvironment. The writing of the manuscript was led by Y.D. with participation from W.W.

## Competing interests

The authors declare no competing interests.
