## [Transparent Peer Review file · Nature Communications]

Long-term Self-driven Metronomic Photodynamic Therapy for Cancer using Alginate Microcapsules Containing Bioluminescent Bacteria and Photosensitizer

Corresponding Author: Professor Zhengwei Mao

Version 0:

Reviewer comments:

Reviewer #1

(Remarks to the Author)

This is an interesting paper describing the preparation of alginate microcapsules containing bioluminescent bacteria and neutral red dye as photosensitizer to allow metronomic PDT after intratumoral injection into mouse melanoma or rabbit VX2 tumors.

- 1 The title of a paper is meant to summarize the contents not to be a mystery teaser. I suggest "Long-term Self-driven Metronomic Photodynamic Therapy for Cancer using Alginate Microcapsules Containing Bioluminescent Bacteria and Photosensitizer"
- 2 Use "bioluminescent bacteria" not "luminous bacteria"; Use "bioluminescence" not "fluorescence"
- 3 The choice of neutral red as a photosensitizer is somewhat puzzling. In one place they talk about also testing chlorin(e6) but I could not find any details of a comparison. Neutral red is a Type I PS (generates hydroxyl radicals) while chlorin(e6) is a Type II PS (generates singlet oxygen). This should be discussed in text at some length.
- 4 Dihydroethidium is a probe for superoxide not really for ROS.
- 5 There is no discussion on the vital role of oxygen in their approach. Firstly the bioluminescence emission from bacteria is itself oxygen dependent. Secondly the PDT effect is oxygen dependent. Therefore their approach would be even more limited by the hypoxic tumor environment than conventional PDT.
- 6 Neutral red should be specified in the abstract and introduction; I spent some time searching for what the PS actually was.
- 7 It should be "Vibrio harveyi" and "Aliivibrio fischeri"
- 8 The power density (2-5 mW/cm²) should be specified in abstract and results. This is one of the most important results
- 9 The concentration of the NR in μM should be specified in the microcapsules, cell culture medium, and animal tumors
- 10 In the cell culture experiments the microcapsules and cells were kept separate in a transwell chamber. This is a very unusual setup for PDT. They must identify which diffusible molecules could pass from one chamber to another
- 11 In Figure 7G there should be the Kaplan-Meier curve for oxaliplatin

Reviewer #2

(Remarks to the Author)

The article presents an innovative photodynamic therapy (PDT) approach that combines bioluminescent bacteria with a photosensitizer, eliminating the need for an external light source and enhancing PDT applicability. Additionally, encapsulating the bioluminescent bacteria in alginate microcapsules improves stability, prevents bacterial leakage, and prolongs light emission. The study confirms the photodynamic effect through reactive oxygen species (ROS) detection and employs metabolomics analysis to investigate the impact of oxidative stress and immune activation on cancer cells. Various analytical techniques, including cell viability assays, flow cytometry, ELISA, and immunofluorescence, were used to verify the chemical and biological mechanisms of self-driven PDT. Finally, both in vitro (cellular level) and in vivo (murine melanoma and rabbit hepatocarcinoma) experiments provided strong validation. I believe this article requires only minor revisions before being suitable for publication in Nature Communications.

1. Does NHS-NR undergo photobleaching under prolonged irradiation? This could affect ROS generation efficiency.
2. Does PB@MC degrade in vivo? Are the degradation products toxic?

3. Have you considered intravenous injection as a delivery method?
4. In Figure 3C, why is there an abrupt change in fluorescence intensity at the beginning of LED irradiation?

Reviewer #3

(Remarks to the Author)

The authors present compelling data on an innovative approach to photodynamic therapy. The PB@MCs utilized are well-characterized and induce strong anti-tumor efficacy across multiple tumor models by inducing the generation of ROS as well as yielding favorable changes to tumor-infiltrating immune cells. The results presented are novel and of importance to the field however, there are a few issues that must be addressed.

First, multiple statistical analyses were performed incorrectly. For all figures where data from more than 2 groups is presented the student's t-test is improper; instead, the authors must use either a one-way ANOVA (parametric) or Kruskal-Wallis (non-parametric) test with the appropriate post-hoc test to assess differences between groups. The authors' conclusion that PB@MCs offer a unique benefit would be strengthened by showing statistical differences between PB@MCs and groups other than the vehicle control, if such differences are present. Additionally, no statistics are shown for the survival plots, despite what appear to be large differences in survival between groups; these analyses should be performed with the results clearly displayed and/or stated in text. Furthermore, in multiple instances differences are described as "significant" when no statistics are presented (Lines 170, 182, 267, 317). The name of the statistical test conducted should be included in the figure legends to improve clarity of the results presented.

Second, while the authors show that PB@MCs do not induce weight loss or noticeable pathologic changes to off-target organs after injection, it would be beneficial to see an assessment of the immunogenicity and longevity of PB@MCs. Do mice or rabbits generate an immune response specific to the PB@MCs? Does such a response preclude the administration of more than one dose? While it was shown PB@MCs can remain bioluminescent for up to 50 hours in vitro, how long are the bacteria able to survive in the PB@MCs? It was stated that "virtually no" bacterial escape was observed in vitro but was this assessed in vivo? Do animals show immune responses against V.H.BB170 that suggests bacterial exposure outside of the microcapsules?

The figure legends should be amended to specify which results are from in vitro or in vivo experiments and how long post-treatment tissues were harvested.

Additionally, there are several wording or labeling issues in the text and figures. Specific comments are below:

Line 68: recommend removing "which are commonly animal models"

Line 72: While data shows efficacy in two models, saying the therapy is "universally applicable" is unsupported and should be reworded.

Lines 84-86: "These bacteria...to DMEM)" this sentence is confusing. As written, it reads as if physiological conditions allowed the cells to live and the cells were exposed to a light source, when the results actually show survival and light emission were measured as outcomes.

Lines 91-94: "For photosensitization...bacteria" Recommend rewording as chlorin e6 was not actually used for any experiments reported here, the spectrum was just used as reference.

Line 122: Is the formula for individual PB@MC a representation of the composition of the average particle? How much variability is present between individual PB@MCs?

Line 132: The phrase "virtually no" implies some degree of bacterial escape occurs. How was this assessed?

Figure 2B: The methods describe the LIVE/DEAD staining as calcein and PI, however in the results text this is described as red fluorescence from EthD-1. Additionally, no quantitative data from the LIVE/DEAD staining is shown in the figure, however, the text specifies 90% cell death. These contradictions should be clarified.

Figure 2D: In text or in figure legend, the markers used to differentiate early and late apoptosis should be specified.

Figure 3B: The description of the assay in the figure legend and text does not match the labels of the panel. What is "BBoxiProbe?"

Line 227: "Glutathione metabolism" does not appear in Figure 4F or SI Fig 13. This should be clarified.

Lines 243-246: Background information on DAMPS and DC maturation should have citations.

Figure 5D: It should be clarified in text or Figure legend that DC maturation was examined in vitro.

Line 265: Figure 5E shows an increase in CD8 T cells but does not assess their cytotoxic potential, recommend removing the word "cytotoxic."

Line 281: From the results presented in Figure 6, it appears all tumors treated with PB@MCs resolved completely, but the stated results say "almost total tumour eradication." If there was a tumor that did not resolve, this data should be included.

Line 294: TUENL should be TUNEL.

Lines 298-299: The data presented shows a reduction in CD206 staining, which could be an overall reduction in macrophage infiltration or a specific reduction in M2-like macrophages. The data shown does not show differentiation to an M1-like phenotype. This is again stated in the discussion (Line 375) and Figure 8, however, the conclusion that PB@MCs induce a transition from M2 to M1 phenotypes is not supported by the data shown.

Lines 320-321: "H&E...metastases" This is an incomplete sentence that reads more like a figure legend heading, suggest rewording.

Line 333: Recommend removing "Or" at the start of the sentence.

Line 363: inhibiting should be inhibited

Line 374: From the data shown, the differentiation of CD4 and CD8 T cells was not assessed. This data should be presented or the sentence should be reworded to avoid confusion.

Line 400: Resolution of hepatocarcinoma tumors in rabbits was not resolved in 2 days by the data shown. Recommend rewording sentence for better clarity.

Lines 544-561. In the methods "Cell Culture" section details on how cells were cultured is included twice with almost identical wording. This duplication should be fixed.

Lines 598-607. The methods description of how ROS were detected in vivo should clarify when and how mice were treated and at what time post-treatment tumors were harvested.

Lines 699-702: Immunostaining for CD4, CD8, CD206, and CD86 was only shown in mouse tissues, the text here implies this was also performed with rabbit tumors.

Figure 2 Legend: The legend should clarify what the colors shown in 2B represent (ie green is live cells and red is dead). (E) Legends state PI staining is shown, but text and figure itself have DAPI staining. This should be clarified.

Figure 4: The title suggests the metabolomics of the PB@MCs themselves are being investigated, when the data actually examines how PB@MCs alter metabolism of tumor cells. This should be clarified. (D) By what criteria were the top 30 metabolites selected: p-value, fold-change, or another metric?

Figure 5 Legend: For panels D-E, it should be specified what the proportion of cells shown are out of (ie % mature DCs out of total DCs, % CD4 of CD45+). Panel F, y-axis includes (%) but a ratio is not a percentage.

Figure 6H: It should be clarified what FITC staining represents.

Figure 7H: It should be clarified what is included in the merge image. Is the blue staining DAPI? (I) Is the outline in the liver panels indicating tumor tissue? Please clarify.

Reviewer #4

(Remarks to the Author)

Version 1:

Reviewer comments:

Reviewer #1

(Remarks to the Author)

1. Give examples of other papers where metabolomics analysis has been used to determine mechanisms of PDT
2. They used "bioluminescent bacteria" in title as requested but then continued to use "bioluminous bacteria" in the remainder. Please correct this throughout
3. "The generation of •OH is independent of oxygen availability" is incorrect. Should be "The generation of •OH is less dependent on oxygen availability"
4. Change "via an electrostatic field" to "via electrostatic attraction"
5. What does "surface opaque melanoma" mean?

Reviewer #2

(Remarks to the Author)

I think authors carefully revised the manuscript according to the comments. I recommend publication of the revised one as is.

Reviewer #3

(Remarks to the Author)

Authors have properly addressed all my comments and have improved the manuscript.

Reviewer #4

(Remarks to the Author)

Reviewer #1 (PDT, engineered bacteria):

This is an interesting paper describing the preparation of alginate microcapsules containing bioluminescent bacteria and neutral red dye as photosensitizer to allow metronomic PDT after intratumoral injection into mouse melanoma or rabbit VX2 tumors.

Response: We thank the reviewer for thinking that our work is interesting. To improve this study, we performed more experiments and added more clarifications according to the reviewer's comments.

1 The title of a paper is meant to summarize the contents not to be a mystery teaser. I suggest "Long-term Self-driven Metronomic Photodynamic Therapy for Cancer using Alginate Microcapsules Containing Bioluminescent Bacteria and Photosensitizer"

Response: We thank the reviewer's comment. We have changed the title of in revised manuscript according to the reviewer's comments.

2 Use "bioluminescent bacteria" not "luminous bacteria"; Use "bioluminescence" not "fluorescence"

Response: Thanks for the reviewer's suggestion. We have carefully go through the full manuscript and corrected the expression of "bioluminescent bacteria" and "bioluminescence". The changes have been highlighted in revised manuscript and an attached file, "Manuscript with highlighted changes".

3 The choice of neutral red as a photosensitizer is somewhat puzzling. In one place they talk about also testing chlorin(e6) but I could not find any details of a comparison.

Neutral red is a Type I PS (generates hydroxyl radicals) while chlorin(e6) is a Type II PS (generates singlet oxygen). This should be discussed in text at some length.

Response: Thanks for the reviewer's suggestion. We have addressed these comments by providing clarifications. Clinical photosensitizers predominantly rely on oxygen-dependent Type II mechanisms, which limit their therapeutic efficacy due to the hypoxic tumor microenvironment. Developing oxygen-independent Type I photosensitizers represents a critical approach for effective photodynamic therapy in hypoxic tumors. Notably, in this study, Neutral Red (NR), a Type I photosensitizer, demonstrated particular suitability for intratumoral photodynamic therapy. Given that the emission spectrum of the *V.H.BBI70* strain showed robust emission within the 400-600 nm range, we compared the UV-Vis absorption spectra of the two photosensitizers to select the most suitable match for bacterial bioluminescence. As shown in **Figure R1**, compared to Ce6-NHS, NR-NHS exhibited superior absorption within the 400-600 nm range, making it more appropriate for our experiment. Additionally, NR functions as a Type I photosensitizer, primarily generating hydroxyl radicals ($\bullet\text{OH}$), whereas Ce6 acts as a Type II photosensitizer, mainly producing singlet oxygen ($^1\text{O}_2$). The generation of hydroxyl radicals is independent of oxygen availability, indicating that NR as a Type I photosensitizer is likely more advantageous in hypoxic tumor regions. We presented the absorption spectra of the photosensitizers in the supplementary materials (SI Fig. 3) and provided a concise explanation in lines 92–103 of the revised manuscript: “For photosensitisation and initiation of ROS generation, N-hydroxysuccinimide neutral red (NHS-NR) and N-hydroxysuccinimide chlorin e6 (NHS-Ce6) photosensitizer are both commonly used PS molecules. Since the *V.H.BBI70* strain emits robust bioluminescence within the 400–600 nm range, we compared the UV–Vis absorption spectra of these two PS molecules to identify the one best matching the bacterial emission range. As shown in **SI Fig. 3**, the emission of *V.H.BBI70* ideally aligns with the absorption range of NHS-NR. Additionally, NR functions as a Type I photosensitizer, primarily generating hydroxyl radicals ($\bullet\text{OH}$), while Ce6 acts as a Type II photosensitizer, mainly generating singlet oxygen ($^1\text{O}_2$). The generation of $\bullet\text{OH}$ is

independent of oxygen availability, suggesting that NR, as a Type I photosensitizer, may offer greater advantages in hypoxic tumor regions.”

Fig. R1 Absorption spectra of NHS-Neutral red and NHS-Ce6 photosensitizers

4 Dihydroethidium is a probe for superoxide not really for ROS.

Response: We agree with the reviewer’s comments. Since Neutral Red is a Type I photosensitizer (PS) generating $\bullet\text{OH}$, we selected a $\bullet\text{OH}$ assay kit (Tissue $\bullet\text{OH}$ Assay Kit, BBoxProbei O28, BB-46072, BestBio, Shanghai, China) (*Theranostics* 11, 379-396 (2021); *Chemical Engineering Journal* 411, (2021); *Biomaterials* 249, 120054 (2020)), to evaluate the $\bullet\text{OH}$ levels produced by PB@MCs in mouse tumor tissues *in vivo*. Four groups of B16 melanoma-bearing mice were injected intratumorally with 50 μL PBS (control), B@MCs, P@MCs, or PB@MCs. After 10 hours, tumors were aseptically collected, weighed, and processed according to the manufacturer’s instructions. Tumor tissues were homogenized by cryo-grinding at 60 Hz for 120 s, then adjusted to 50 mg/mL with sterile PBS. Subsequently, 190 μL of tumor homogenate (50 mg/mL) and 10 μL of BBoxiProbe O28 working solution were added to a 96-well black plate and incubated at 37°C in the dark for 20 min. Fluorescence intensity was measured using a microplate reader (excitation: 488 nm; emission: 520 nm). Protein concentrations in the homogenate were quantified using the Bradford Protein Assay Kit

(P0006, Beyotime Biotechnology, Shanghai, China). Tissue \bullet OH levels were expressed as fluorescence intensity normalized by protein concentration. As shown in **Figure R2**, mice treated with PB@MCs exhibited significantly higher fluorescence intensity compared to the other groups, demonstrating enhanced ROS (\bullet OH) generation and confirming the *in vitro* findings.

Fig. R2 Characterisation of ROS generation efficiency *In Vivo* by PB@MCs.

In vivo detection of ROS content. ROS production was assessed using the Tissue hydroxyl radicals Assay Kit (BBoxiProbe O28 probe). For comparison, B16 tumor-bearing mice of either treated by 50 μ L PBS (control group), Empty MCs linked to photosensitizer (P@MCs, contain 0.055 μ mol NR), MCs encapsulating BB170 bacteria (B@MCs), or PB@MCs for 10 h. 190 μ L of B16 tumor homogenate (50 mg/mL) and 10 μ L of BBoxiProbe O28 working solution were added to a 96 black well plate, incubated at 37°C in darkness for 20 minutes, and fluorescence intensity was detected by a Microplate Reader (excitation at 488 nm, emission at 520 nm) (n = 3 biologically independent samples). The data are presented as mean \pm SD. Statistical significance is noted with ****p < 0.0001, compared to the data for P@MCs, B@MCs and control group according to one-way ANOVA test combination with Tukey's multiple comparisons by GraphPad Prism 9 XML project software.

We have made the corresponding modifications in lines 211-216 of revised manuscript, as “Following the experiment, tumors were aseptically harvested, weighed, and processed according to the Tissue \bullet OH assay kit (BBoxiProbe O28 probes) to evaluate

the *in vivo* •OH levels. As shown in **Fig. 3B**, mice treated with PB@MCs exhibited significantly higher fluorescence intensity compared to the other groups, demonstrating enhanced •OH generation and confirming the *in vitro* findings.” and in lines 661-673 as “To evaluate the •OH levels produced by PB@MCs in mouse tumor tissues *in vivo*. Four groups of B16 melanoma-bearing mice were injected intratumorally with 50 μ L PBS (control), B@MCs, P@MCs, or PB@MCs. After 10 hours, tumors were aseptically collected, weighed, and processed according to the manufacturer’s instructions. Tumor tissues were homogenized by cryo-grinding at 60 Hz for 120 s, then adjusted to 50 mg/mL with sterile PBS. Subsequently, 190 μ L of tumor homogenate (50 mg/mL) and 10 μ L of BBoxiProbe O28 working solution were added to a 96-well black plate and incubated at 37°C in the dark for 20 min. Fluorescence intensity was measured using a microplate reader (excitation: 488 nm; emission: 520 nm). Protein concentrations in the homogenate were quantified using the Bradford Protein Assay Kit (P0006, Beyotime Biotechnology, Shanghai, China). Tissue •OH levels were expressed as fluorescence intensity normalized by protein concentration.” All changes have highlighted the revised portions.

5 There is no discussion on the vital role of oxygen in their approach. Firstly the bioluminescence emission from bacteria is itself oxygen dependent. Secondly the PDT effect is oxygen dependent. Therefore their approach would be even more limited by the hypoxic tumor environment than conventional PDT.

Response: We agree with the reviewer’s comments. Bacterial bioluminescence intrinsically depends on oxygen. To minimize oxygen dependency, we employed Neutral Red (NR) as a Type I photosensitizer. However, given the hypoxic nature of tumor microenvironments, evaluating the bioluminescence performance and duration of PB@MCs under these conditions is essential. Thus, we designed an experiment to investigate the bioluminescence properties of PB@MCs within solid hypoxic tumors. Using an IVIS imaging system, bioluminescence duration was evaluated in B16

melanoma-bearing mice (tumor volume approximately 300 mm³) following intratumoral injection of 50 µL PB@MCs. Imaging was conducted at 0, 2, 4, 6, 8, and 10 hours post-injection. As shown in **Figure R3**, PB@MCs exhibited sustained bioluminescent signals in solid tumors for at least 10 hours, confirming their bioluminescent capability within hypoxic tumor conditions. However, the bioluminescent signals of PB@MCs gradually attenuated over time, with weakening first observed at the tumor core, progressively extending towards the periphery. As noted by the reviewer, this phenomenon may result from a hypoxic gradient within tumors, where oxygen deprivation intensifies towards the core, whereas peripheral regions remain relatively oxygenated. Consequently, stronger bioluminescent signals were detected in peripheral areas of tumors. The bioluminescence duration of PB@MCs *in vivo* was shorter than the 50-hour duration observed *in vitro*, possibly due to signal intensity falling below the IVIS system's detection limit (potentially influenced by melanin absorption in melanoma and limited tissue penetration depth etc.), or more likely due to ongoing oxygen consumption during sustained bacterial bioluminescence within tumors. Following the experiment, tumors were aseptically harvested, weighed, and processed according to the Tissue •OH assay kit's instructions. The resulting tissue •OH levels are presented in **Figure R2**.

Fig. R3 Bioluminescence image (BLI) of B16 melanoma-bearing mice injected with PB@MCs.

B16 melanoma-bearing mice with tumor volumes $\sim 300 \text{ mm}^3$ were intratumorally injected with $50 \mu\text{L}$ of PB@MCs. Bioluminescence imaging was performed using an IVIS imaging system (PerkinElmer, IVIS Lumina Series III) at 0, 2, 4, 6, 8, and 10 hours post-injection ($n = 3$ biologically independent samples). All pictures were analysis by Living image 4.8.0 software.

We have added the Figure into SI information as **SI Fig.13** and made the corresponding modifications in lines 196-211 of the revised manuscript: “We also evaluate the bioluminescence performance and duration of PB@MCs in mice. As shown in **SI Fig.13**, PB@MCs exhibited bioluminescent signals in tumors for at least 10 hours, confirming their bioluminescent capability within hypoxic tumor microenvironments. However, the bioluminescent signals of PB@MCs gradually attenuated over time, with the signal weakening initiating from the tumor core and progressively diminishing toward the periphery. this phenomenon may be attributed to the hypoxic gradient within

tumors, where oxygen deprivation intensifies toward the core while relatively oxygenated conditions prevail in peripheral regions. Consequently, stronger self-luminescent signals were observed in areas distal to the tumor center. The bioluminescence duration of PB@MCs *in vivo* was shorter than the 50-hour duration observed *in vitro*, possibly due to signal intensity falling below the IVIS system's detection limit (potentially influenced by melanin absorption in melanoma and limited tissue penetration depth etc.), or more likely due to ongoing oxygen consumption during sustained bacterial bioluminescence within tumors." All changes have highlighted the revised portions.

6 Neutral red should be specified in the abstract and introduction; I spent some time searching for what the PS actually was.

Response: We thank the reviewer's comment. we added a short specified explanation of Neutral red photosensitizer we used in abstract as "In this study, we integrated self-bioluminescent bacteria with a neutral red (NR) photosensitizer in alginate microcapsules (MCs) to create a self-driven metronomic photodynamic therapy (Sd-PDT) system that can be securely implanted within a tumour" and introduction in lines 60-62 of the revised manuscript, as "Here, we combined bioluminous bacteria and Neutral red (NR) photosensitizer within alginate microcapsules (MCs) to develop the first self-driven metronomic photodynamic system (Sd-PDT) for cancer therapy." The changes have been highlighted in an attached file, "Manuscript with highlighted changes".

7 It should be "Vibrio harveyi" and "Aliivibrio fischeri"

Response: Thanks for the reviewer's suggestion. We have go through the full manuscript and corrected the expression of "Vibrio harveyi" and "Aliivibrio fischeri". The changes have been highlighted in an attached file, "Manuscript with highlighted changes".

8 The power density (2-5 mW/cm²) should be specified in abstract and results. This is one of the most important results

Response: Thanks for the reviewer's suggestion. we added a short explanation on the results of the power density (1-5 mW/cm²) in abstract as "By harnessing nutrients from the tumour microenvironment, this system sustains the ~50 hours long lasting bioluminescence with the power density (1-5 mW/cm²) and generation of reactive oxygen species (ROS) than that of a standard LED radiation." and results in lines 157-162 of the revised manuscript, as "The laser power meter was used to plot the linear relationship between the bioluminescence intensity (10⁴ cps) and the power density (mW/cm²) of PB@MCs (SI Fig. 8), and further calculations reveal that PB@MCs can express an power density of 1-5 mW/cm² in DMEM medium for up to 50 h (Fig. 1H), which is suitable for mPDT photoirradiation dose and significantly surpasses the duration of all currently available chemical-driven PDT. "The changes have been highlighted in an attached file, "Manuscript with highlighted changes".

9 The concentration of the NR in μ M should be specified in the microcapsules, cell culture medium, and animal tumors

Response: We thank the reviewer's suggestions. We have defined the specified concentration of the NR in the revised manuscript, including microcapsules, cell culture medium, and animal tumors, and other experimental groups involving NR group. The changes have been highlighted in an attached file, "Manuscript with highlighted changes".

10 In the cell culture experiments the microcapsules and cells were kept separate in a transwell chamber. This is a very unusual setup for PDT. They must identify which diffusible molecules could pass from one chamber to another

Response: To address the reviewer's concern, we performed additional experiments. First, we employed DCFH (CAS: 106070-31-9, Biofount Beijing Bio-Tech Co., Ltd.), a commonly used ROS probe, to detect ROS levels in both upper and lower chambers of the 24-well Transwell system. PB@MCs (200 μ L) were incubated in the upper chamber with phenol red-free DMEM medium at 37°C in darkness for 2 hours. Subsequently, 100 μ L of medium from each chamber containing DCFH working solution (40 μ M) was incubated in a 96-well black plate at 37°C in darkness for 20 minutes. Fluorescence intensity was measured using a microplate reader (excitation at 488 nm, emission at 520 nm). As shown in **Figure R4**, both chambers exhibited significant fluorescence intensity, indicating that ROS diffused into the lower chamber. Second, since Neutral Red (NR) is a Type I photosensitizer that generates hydroxyl radicals, we repeated this experiment using a hydroxyl radical probe (Hydroxyphenyl fluorescein, HPF, MX4805, CAS: 359010-69-8 BestBio). Similar procedures were followed: PB@MCs (200 μ L) were incubated in the upper chamber in phenol red-free DMEM medium at 37°C in darkness for 2 hours, after which 100 μ L of medium from each chamber containing HPF working solution (10 μ M) was incubated in darkness for 20 minutes. Fluorescence intensity was measured (excitation at 492 nm, emission at 515 nm). As presented in **Figure R4**, both chambers again showed significant fluorescence intensity, with the upper chamber showing slightly higher fluorescence, consistent with the DCFH results, further indicating ROS diffusion into the lower chamber. These experimental results strongly support the confocal microscopy observations of ROS generation in HepG3B and A375 cells, as depicted in Figure 3A and SI Fig. 12 of the manuscript.

Fig. R4 ROS express levels in upper and lower chamber of 24-transwell model with PB@MCs.

200 μ L of PB@MC were incubated in the upper chamber in darkness at 37°C for 2 hours in phenol red-free DMEM medium in 24-transwell model. After 2hours, 100 μ L medim from upper and lower chamber containing DCFH (40 μ M) or HPD (10 μ M) wording solution and further incubated in darkness in 96 well-black plate for another 20 minutes, and fluorecence intensity was detected using a Microplate Reader (excitation at 488 nm, emission at 520 nm for DCF) and (excitation at 492 nm, emission at 515 nm for HPF). The data are presented as mean \pm SD (n = 3). Statistical significance is noted with *p < 0.05, upper chamber group compared to the lower chamber group according to two tail t-test by GraphPad Prism 9 XML project software.

11 In Figure 7G there should be the Kaplan-Meier curve for oxaliplatin

Response: We thank the reviewer's suggestions. The Kaplan-Meier curve for oxaliplatin were added in Figure 7G of the revised manuscript.

Reviewer #2 (PDT, nanomedicine):

The article presents an innovative photodynamic therapy (PDT) approach that combines bioluminescent bacteria with a photosensitizer, eliminating the need for an external light source and enhancing PDT applicability. Additionally, encapsulating the bioluminescent bacteria in alginate microcapsules improves stability, prevents bacterial leakage, and prolongs light emission. The study confirms the photodynamic effect through reactive oxygen species (ROS) detection and employs metabolomics analysis

to investigate the impact of oxidative stress and immune activation on cancer cells. Various analytical techniques, including cell viability assays, flow cytometry, ELISA, and immunofluorescence, were used to verify the chemical and biological mechanisms of self-driven PDT. Finally, both in vitro (cellular level) and in vivo (murine melanoma and rabbit hepatocarcinoma) experiments provided strong validation. I believe this article requires only minor revisions before being suitable for publication in Nature Communications.

Response: We thank the reviewer for thinking that our work is suitable for publication in Nature Communications. To improve this study, we performed more experiments and added more clarifications according to the reviewer's comments.

1. Does NHS-NR undergo photobleaching under prolonged irradiation? This could affect ROS generation efficiency.

Response: We thank the reviewer's suggestions. To accommodate the reviewer's concern, we performed additional experiments to determine whether the photosensitizer undergoes photobleaching under prolonged irradiation, potentially affecting ROS generation efficiency. The photosensitizer NHS-NR (100 µg/mL) was continuously irradiated for 50 hours using a blue LED light (5 mW/cm²). As shown in **Figure R5(A)**, visible fading of the NR solution color occurred after irradiation. Furthermore, we assessed ROS generation efficiency using DCFH (CAS: 106070-31-9, Biofount Beijing Bio-Tech Co., Ltd.) before and after 50 hours of irradiation. As demonstrated in **Figure R5(B)**, the fluorescence intensity significantly decreased, but approximately 58% of the initial ROS production efficiency remained. These findings suggest that although the NR photosensitizer undergoes photobleaching following 50 hours of continuous irradiation, it still retains substantial functionality in ROS generation, despite reduced efficiency.

Fig. R5 Photobleaching test of NHS-NR in 50 h.

The photosensitizer NHS-NR (100 $\mu\text{g}/\text{mL}$) was continuously irradiated for 50 hours under blue LED light (5 mW/cm^2). (A) A photo picture was taken before and after the reaction. (B) At 0 h and 50 h, DCFH (40 μM) working solution and further incubated in darkness in 96 well-black plate for another 20 minutes, and fluorescence intensity was detected using a Microplate Reader (excitation at 488 nm, emission at 520 nm). The data are presented as mean \pm SD ($n = 3$). Statistical significance is noted with *** $p < 0.001$, compared to the 50 h group according to two tail t-test by GraphPad Prism 9 XML project software.

2. Does PB@MC degrade in vivo? Are the degradation products toxic?

Response: Thanks for asking this question. We made additional experiments to accommodate the reviewer's concern. Three healthy C57BL/6 mice (aged 6–8 weeks, weighing approximately 20 g) received subcutaneous injection of 50 μL PB@MC suspensions. Mice were euthanized on days 0, 1 and 3 post-injection to collect PB@MCs and blood samples. As shown in **Figure R6**, PB@MCs recovered at these intervals retained structural integrity without significant bacterial leakage, indicating that no substantial degradation occurred over the 3 day observation period. These findings align with our previous study (*Nat Commun* 13, 4495 (2022)), demonstrating that calcium alginate microspheres maintain structural stability *in vivo*. As the same comments asked by reviewer 3. White blood cell count (WBC) and procalcitonin (PCT) are commonly used inflammatory markers following bacterial infection. We injected

PBS (control), PB@MCs, or empty microspheres (MCs) (3 mice per group, 50 μ L) into healthy mice. Blood test analysis performed on days 1, 3, and 7 revealed no significant abnormalities in WBC levels across all groups within the 7 day period (normal range $0.8\sim 6.8 \times 10^9/L$, Materials Today 51, 96-107 (2021)) (**Figure R7**). Serum PCT levels measured on days 1, 3, and 7 showed that, compared to the control group, the PB@MCs group exhibited elevated PCT on day 1. PCT levels in all groups decreased to low levels by day 3 and 7. These findings suggest that PB@MCs may induce a degree of inflammation, likely attributable to their generation of ROS, but this response gradually subsides after 3 days. Consequently, PB@MCs are optimally suited for sustained PDT aimed specifically at tumor tissues. We have added the Figure into SI information as **SI Fig.18** and made the corresponding modifications in lines 336-343 of the revised manuscript.

Fig. R6 Microscopy images of PB@MCs *in vivo*.

3 healthy C57b6 mice were intramuscular injections of PB@MCs suspensions (50 μ L). Mice were euthanized on days 0, 1, and 3 to procure PB@MCs.

Fig. R7 Blood test of WBC and PCT in healthy mice

Blood was collected from 8-week-old C57B6 mice by eyeball blood collection on day 1, 3 and 7 after treatment, including injection of 50 μ L PBS(control), PB@MCs, and MCs.

3. Have you considered intravenous injection as a delivery method?

Response: We thank the reviewer for raising this important question. Based on our experimental findings and design, PB@MCs typically have a particle size ranging from 100 to 150 μ m. Such particle sizes pose risks of vascular occlusion, potentially compromising therapeutic efficacy if administered intravenously. Consequently, PB@MCs microspheres are more suitable for localized administration methods, such as percutaneous intratumoral injection or transarterial chemoembolization (TACE), enabling direct targeting of tumor sites.

4. In Figure 3C, why is there an abrupt change in fluorescence intensity at the beginning of LED irradiation?

Response: We thank the reviewer for raising this important question. Figure 3C illustrates the duration and intensity of ROS generated by our developed Self-driven Metronomic Photodynamic System (Sd-PDT) compared with conventional PDT (external LED irradiation) within a single treatment session. Additionally, the total ROS production between the two modalities under identical therapeutic conditions was compared. In the conventional PDT protocol, Neutral Red (NR) was irradiated with an LED light source (300 mW/cm^2) for 1 hour per session. For the Sd-PDT system, PB@MC microspheres loaded with an equivalent dose of NR were employed. ROS fluorescence intensity was quantified using the DCFH probe. As shown in **Figure 3C**, the LED-irradiated group exhibited a rapid increase in ROS fluorescence during irradiation, indicating faster and stronger initial ROS generation. However, ROS production ceased immediately upon LED termination. In contrast, although the PB@MC group initially showed lower fluorescence intensity, it maintained continuous ROS generation throughout the treatment period. These results confirm that the Sd-PDT system enables sustained ROS production following a single treatment, demonstrating superior prolonged therapeutic activity compared to conventional external light-activated PDT in a single treatment session. These findings have been included in the revised manuscript as Figures 3C and 3D.

To accommodate the reviewer's concern, we added more explanation of Figure 3C in lines 216-232 of the revised manuscript, as "To verify that the cumulative ROS production depends on the bioluminescence intensity and exposure duration of PB@MCs, ROS generation was continuously monitored for approximately 50 hours. The duration and intensity of ROS generated by Sd-PDT were compared with those of conventional PDT (external LED at 300 mW/cm^2 irradiating NR for 1 hour per session). For the Sd-PDT group, PB@MC microspheres loaded with an equivalent NR dose were

used. ROS fluorescence intensity was quantified using the DCFH probe at each time point. As shown in **Figure 3C**, the LED group exhibited a significant increase in ROS fluorescence signals during the 1-hour irradiation, indicating rapid and strong initial ROS generation. However, ROS production ceased immediately after irradiation ended. Conversely, the PB@MC group displayed persistent ROS generation throughout the treatment session (~50 h), despite lower initial fluorescence intensity compared to the LED group. These results confirm the sustained ROS production of the Sd-PDT system in a single treatment session, reflecting their persistent effectiveness in the tumor environment. The cumulative ROS quantity in the PB@MC group significantly exceeded that of the LED group (**Figure 3D**). These findings provide substantial evidence of the durable ROS-generating capacity of PB@MCs.”

Reviewer #3 (Preclinical models, immune readouts):

The authors present compelling data on an Innovative approach to photodynamic therapy. The PB@MCs utilized are well-characterized and induce strong anti-tumor efficacy across multiple tumors models by inducing the generation of ROS as well as yielding favorable changes to tumor-infiltrating immune cells. The results presented are novel and of importance to the field however, there are a few issues that must be addressed.

Response: We thank the reviewer for thinking that our work is are novel and of importance to the field. To improve this study, we performed more experiments and added more clarifications according to the reviewer's comments.

1. First, multiple statistical analyses were performed incorrectly. For all figures where data from more than 2 groups is presented the students t-test is improper; instead, the authors must use either a one-way ANOVA (parametric) or Kruskal-Wallis (non-parametric) test with the appropriate post-hoc test to assess differences between groups. The authors conclusion that PB@MCs offer a unique benefit would be strengthened by

showing statistical differences between PB@MCs and groups other than the vehicle control, if such differences are present. Additionally, no statistics are shown for the survival plots, despite what appear to be large differences in survival between groups; these analyses should be performed with the results clearly displayed and/or stated in text. Furthermore, in multiple instances differences are describes as “significant” when no statistics are presented (Lines 170, 182, 267, 317). The name of the statistical test conducted should be included in the figure legends to improve clarity of the results presented.

Response: We thank the reviewer for this suggestion. We have carefully reviewed the entire manuscript and corrected the statistical analyses for all figures involving more than two groups, employing one-way ANOVA with appropriate post-hoc tests to assess differences among groups. Additionally, we have presented statistical differences between the PB@MCs and other treatment groups, beyond comparisons with the vehicle control. Statistical analyses for survival plots in Figures 6 and 7 were clearly indicated in the figure legends. All statistical calculations were performed using GraphPad Prism 9 software. The revisions are highlighted in the attached document, “Manuscript with highlighted changes”.

2. Second, while the authors show that PB@MCs do not induce weight loss or noticeable pathologic changes to off target organs after injection, it would be beneficial to see an assessment of the immunogenicity and longevity of PB@MCs.

Do mice or rabbits generate an immune response specific to the PB@MCs?

Thanks for the reviewer’s suggestion. White blood cell count (WBC), and procalcitonin (PCT) are commonly used inflammatory markers following bacterial infection [1]. We subcutaneously injected PBS (control), PB@MCs, or empty microspheres (MCs) (3 mice per group, 50 μ L) into healthy mice. Blood test analysis performed on days 1, 3, and 7 revealed no significant abnormalities in WBC levels across all groups within the 7 day period (normal range 0.8~6.8 $\times 10^9$ /L, Materials Today 51, 96-107 (2021))

(Figure R7). Serum PCT levels measured on days 1, 3, and 7 showed that, compared to the Control group, the PB@MCs group exhibited elevated PCT on day 1. PCT levels in all groups decreased to low levels by day 3 and 7. These findings suggest that PB@MCs may induce a degree of inflammation, likely attributable to their generation of ROS, but this response gradually subsides after 3 days. We have added the Figure into SI information as SI Fig.18 and made the corresponding modifications in lines 336-343 of the revised manuscript.

Fig. R7 Blood test of WBC and PCT in healthy mice

Blood was collected from 8-week-old C57B6 mice by eyeball blood collection on day 1, 3 and 7 after treatment, including injection of 50 μ L PBS (control), PB@MCs, and MCs.

The remarkable upsurge in proinflammatory factors ordinarily signifies immune activation and likely inflammation. IL-6 enhances host defense by stimulating acute

phase responses, hematopoiesis, and immune reactions [2]. The established role of ROS in promoting IL-6 expression[3,4] and downstream immune activation. Considering PB@MCs likely attributable to their generation of ROS, we injected PBS, PB@MCs, and MCs subcutaneously into healthy mice (3 mice per group, 50 μ L). After 1, 3, 7 days, the IL-6 levels in the mouse serum were measured. As illustrated in **Figure R8**, significant changes in IL-6 were observed in Day 1 and Day 3, but absence of significant IL-6 variations across groups in Day 7. These results demonstrate that PB@MCs could elicit immune responses *in vivo*, attributable to its ROS production ability.

Fig. R8 IL-6 secretion levels in healthy mice blood serum following various treatments

[1] Pourakbari B, Mamishi S, Zafari J, et al. Evaluation of procalcitonin and neopterin level in serum of patients with acute bacterial infection. *Braz J Infect Dis.* 2010;14(3):252-255.

[2] Tanaka T, Narazaki M, Kishimoto T. IL-6 in inflammation, immunity, and disease. *Cold Spring Harb Perspect Biol.* 2014;6(10):a016295.

[3] Sah DK, Arjunan A, Park SY, et al. Sulforaphane Inhibits IL-1 β -Induced IL-6 by Suppressing ROS Production, AP-1, and STAT3 in Colorectal Cancer HT-29 Cells. *Antioxidants.* 2024; 13(4):406.

[4] Liu J, Liu Y, Chen J, et al. The ROS-mediated activation of IL-6/STAT3 signaling pathway is involved in the 27-hydroxycholesterol-induced cellular senescence in nerve cells. *Toxicol In Vitro*. 2017;45(Pt 1):10-18.

Does such a response preclude the administration of more than one dose?

In this study, our *in vivo* experiments demonstrated that a single administration of PB@MCs effectively eradicated tumors. Although PB@MCs encapsulate bacteria effectively and maintain structural stability, their prolonged presence in normal tissues as ROS-generating "microfactories" induces inflammatory responses. Thus, this material should be strategically confined to tumor tissues, where it exerts antitumor effects through three coordinated mechanisms: photodynamic therapy (PDT), nutrient deprivation, and pro-inflammatory modulation. The demonstrated efficacy of single-dose tumor eradication renders repeated administrations unnecessary.

While it was shown PB@MCs can remain bioluminescent for up to 50 hours *in vitro*, how long are the bacteria able to survive in the PB@MCs?

To determine the duration of bacterial survival within PB@MCs, we conducted additional monitoring experiments. PB@MCs (50 μ L) were initially cultured in 2216e medium until reaching the stationary phase, after which the medium was replaced with phenol red-free DMEM supplemented with 10% FBS and without antibiotics. Bacterial viability was monitored by measuring the optical density at 600 nm (OD₆₀₀) using a microplate reader. The culture medium was systematically refreshed every 10 hours over a 75-hour observation period. As depicted in **Figure R9**, bacteria encapsulated in PB@MCs remained viable for at least 75 hours, demonstrating sustained survival with >75% viability retention, despite a gradual decline in OD over time.

Fig. R9 Bacteria viability in PB@MCs.

Growth curves for PB@MCs cultured in DMEM medium (phenol red-free, 10% FBS, no antibiotics) at 37 °C for 75 h. PB@MCs (50 μ L) were gathered within 96-well transparent plates, and their optical density at 600 nm was measured using a microplate reader (n = 4).

It was stated that “virtually no” bacterial escape was observed *in vitro* but was this assessed *in vivo*?

To address the reviewer’s concern, we conducted additional experiments to evaluate bacterial escape *in vivo*. B16 melanoma-bearing mice received intratumoral injections of either 50 μ L of V.H.BB170 bacterial suspension (centrifuged bacterial pellet) or PB@MCs. After 24 hours, blood samples and tissue specimens (liver, spleen, lung, kidney, and tumor) were collected aseptically from the mice. Tumor tissues containing PB@MCs were excluded to prevent microsphere rupture during mechanical homogenization, which could lead to false-positive bacterial escape results. Tissues were weighed and homogenized in sterile PBS to achieve 50 mg/mL suspensions. Each homogenate tube received three grinding beads and underwent mechanical disruption at 60 Hz for 120 s using a cryogenic mill. After centrifugation at 1,000 \times g for 1 min, 100 μ L aliquots of supernatant were plated. Blood samples were diluted 20-fold in PBS, vortexed thoroughly, and plated in 100 μ L aliquots. All plates were incubated at 37°C for 24 hours, then photographed under ambient light and in darkness. As shown in **Figure R10**, mice treated with non-encapsulated bacteria exhibited systemic bacterial

dissemination to multiple organs, with detectable bioluminescent signals observed in the spleen, kidneys, and lungs. As expected, significant bioluminescence was also detected in tumor tissues in V.H.BB170 bacterial suspension group. In contrast, mice treated with PB@MC showed no detectable bioluminescence in any organs except tumors, where only limited signals were observed. These results demonstrate that PB@MCs effectively encapsulate bacteria, resulting in limited bacterial escape from the microspheres.

Fig. R10 Representative photos of bacterial colonies in multiple tissues of tumor-bearing mice.

B16 melanoma-bearing mice were respectively injected intra-tumorally with 50 μ L of V.H.BB170 bacteria, and PB@MCs. After 24 hours, blood samples and tissue specimens including liver, spleen, lung, kidney, tumor were collected from mice under aseptic conditions. Tissues were weighed and homogenized in sterile PBS to achieve 50 mg/mL suspensions. Each homogenate tube received three grinding beads and underwent mechanical disruption at 60 Hz for 120 s using a cryogenic mill. Following centrifugation at 1,000 \times g for 1 min, 100 μ L supernatant aliquots were plated. Blood samples were diluted 20-fold with PBS, vortex-mixed thoroughly, and 100 μ L aliquots plated. All plates underwent 24 hour incubation at 37°C and taking photos in ambient light and darkness.

Do animals show immune responses against V.H.BB170 that suggests bacterial exposure outside of the microcapsules?

We thank the reviewer for their suggestions. In response, we specifically compared the V.H.BB170 group (free bacteria) with the PB@MCs group. We subcutaneously injected PBS (control), V.H.BB170, PB@MCs, or MCs into healthy mice (3 mice per group, 50 μ L). Blood test on days 1, 3, and 7 revealed that the V.H.BB170 group

exhibited a significant increase in WBC on day 3 compared to the other groups. In contrast, the PB@MCs group showed no significant abnormalities in WBC throughout the 7-day period (**Figure R11**). Measurement of PCT levels on days 1, 3, and 7 showed that, compared to the control group V.H.BB170 displayed significantly elevated PCT on days 1 and 3. The PB@MCs group also showed elevated PCT on day 1, but this increase was lower than that in the V.H.BB170 group. PCT levels in all groups decreased to low levels on day 7. These findings indicate that while PB@MCs may induce a degree of inflammation, likely attributable to their generation of ROS, the magnitude of this inflammatory response is significantly lower than that caused by V.H.BB170 free bacteria. Collectively, these results demonstrate that PB@MCs exhibit effective encapsulation of the bacteria.

Fig. R11 Blood test of WBC and PCT in healthy mice

Blood was collected from 8-week-old C57B6 mice by eyeball blood collection on day 1, 3 and 7 after treatment, including injection of 50 μ L PBS (control), PB@MCs, MCs and V.H.BB170 bacteria.

3. The figure legends should be amended to specify which results are from in vitro or in vivo experiments and how long post-treatment tissues were harvested.

Response: We thank the reviewer's suggestions. We have defined the specified which results are from in vitro or in vivo experiments and how long post-treatment tissues were harvested in the figure legends in revised manuscript. The changes have been highlighted in an attached file, "Manuscript with highlighted changes"

4. Line 68: recommend removing "which are commonly animal models"

Response: We thank the reviewer's suggestion. We have removed these words in revised manuscript.

5. Line 72: While data shows efficacy in two models, saying the therapy is "universally applicable" is unsupported and should be reworded.

Response: We thank the reviewer's suggestion. We agree with the reviewer and we have reworded these words in lines 74 of the revised manuscript, as "This study highlights the potential of Sd-PDT as a novel therapeutic strategy applicable to certain tumors, offering promising clinical prospects."

6. Lines 84-86: "These bacteria...to DMEM)" this sentence is confusing. As written, it reads as if physiological conditions allowed the cells to live and the cells were exposed to a light source, when the results actually show survival and light emission were measured as outcomes.

Response: We thank the reviewer's suggestion. We have reworded the sentence in lines 86 of the revised manuscript, as "These three bacterial strains were evaluated for viability and bioluminescent activity under physiological conditions."

7. Lines 91-94: “For photosensitization...bacteria” Recommend rewording as chlorin e6 was not actually used for any experiments reported here, the spectrum was just used as reference.

Response: We thank the reviewer's suggestion. We have reworded the sentence in lines 92-103 of the revised manuscript, as “For photosensitisation and initiation of ROS generation, N-hydroxysuccinimide neutral red (NHS-NR) and N-hydroxysuccinimide chlorin e6 (NHS-Ce6) photosensitizer are both commonly used PS molecules. Since the *V.H.BBI70* strain emits robust bioluminescence within the 400–600 nm range, we compared the UV–Vis absorption spectra of these two PS molecules to identify the one best matching the bacterial emission range. As shown in **SI Fig. 3**, the emission of *V.H.BBI70* ideally aligns with the absorption range of NHS-NR. Additionally, NR functions as a Type I photosensitizer, primarily generating hydroxyl radicals ($\bullet\text{OH}$), while Ce6 acts as a Type II photosensitizer, mainly generating singlet oxygen ($^1\text{O}_2$). The generation of $\bullet\text{OH}$ is independent of oxygen availability, suggesting that NR, as a Type I photosensitizer, may offer greater advantages in hypoxic tumor regions.”

8. Line 122: Is the formula for individual PB@MC a representation of the composition of the average particle? How much variability is present between individual PB@MCs?

Response: We agree with the reviewer's suggestion, the formula for PB@MC representation of the composition of the average particle. We cannot calculate the variability between individual PB@MCs. We have reworded the sentence in lines 129 of the revised manuscript, as “Upon calculation, the formula for the average particle of PB@MC (1×10^3 bacteria cells and 0.028 μg of PS per MC) was established.”

9. Line 132: The phrase “virtually no” implies some degree of bacterial escape occurs. How was this assessed?

Response: Thanks for asking this question. Confocal microscopy analysis (Fig. 1E) were used to revealed no obviously bacterial leakage from the MCs which not have obvious bioluminescence signals. We have reworded the sentence in lines 138-141 of the revised manuscript, as “Confocal microscopy analysis demonstrated that bacterial cells within the MCs self-assembled into structurally stable aggregates while maintaining sustained bioluminescent activity, with no detectable bacterial bioluminescence signals outside the MCs (Fig. 1E).”

10. Figure 2B: The methods describe the LIVE/DEAD staining as calcein and PI, however in the results text this is described as red fluorescence from EthD-1. Additionally, no quantitative data from the LIVE/DEAD staining is shown in the figure, however, the text specifies 90% cell death. These contradictions should be clarified.

Response: We thank the reviewer's suggestion. Cell viability was assessed using the LIVE/DEAD staining assay (Calcein/PI Assay Kit). We have clarified LIVE/DEAD staining and reworded the sentence in lines 170-173 of the revised manuscript, as “Cell viability was assessed using the LIVE/DEAD staining assay (Calcein/PI Assay Kit) and visualised using CLSM. After 8 h, a significant red signal increase in cell mortality was observed in all cell lines in the PB@MC group suggesting a potent cytotoxic effect.”

11. Figure 2D: In text or in figure legend, the markers used to differentiate early and late apoptosis should be specified.

Response: We thank the reviewer's suggestion. We added more clarifications according to the reviewer's comments in line 178-182 as “Apoptosis induced by PB@MCs was investigated using flow cytometry (Figs. 2D, 2E, SI Fig. 11), Cells stained with Annexin V-FITC(+) and DAPI(-) are defined as early apoptotic cells, while cells stained with Annexin-FITC(+) and DAPI(+) are defined as late apoptotic cells.” and corrected the figure legends of Figure 2D in revised manuscript.

12. Figure 3B: The description of the assay in the figure legend and text does not match the labels of the panel. What is “BBoxiProbe?”

Response: Thanks to the reviewer for spotting this issue. BBoxiProbe is the name of Tissue hydroxyl radicals Assay Kit (BBoxProbei O28, BB-46072, BestBio, Shanghai, China). Since NR is a Type I PS generating hydroxyl radicals (\bullet OH). Therefore, we selected a \bullet OH assay kit for *in vivo* detection (*Theranostics* 11, 379-396 (2021); *Chemical Engineering Journal* 411, (2021); *Biomaterials* 249, 120054 (2020)) to evaluate the \bullet OH levels produced by PB@MCs in mice tumor tissues. We have corrected the word and Figure lengend in revised manuscript.

13. Line 227: “Glutathione metabolism” does not appear in Figure 4F or SI Fig 13. This should be clarified.

Response: We thank the reviewers for finding this issues. We have clarified the description in revised manuscript in line 270-275 as “Further analysis revealed that metabolic pathways closely associated with glutathione metabolism, including cysteine and methionine metabolism, biosynthesis of cofactors, ABC transporters, and thyroid hormone synthesis, were significantly suppressed following PB@MC treatment, contributing to increased ROS stress and antitumor effects.” As shown in **Figure R12**, glutathione metabolism is closely linked to pathways such as cysteine and methionine metabolism, biosynthesis of cofactors, ABC transporters, and thyroid hormone synthesis, according to KEGG pathway analysis. Figures 4F and SI Fig. 14 demonstrate that these pathways were significantly suppressed by PB@MC treatment. Thus, PB@MCs may exert regulatory effects on glutathione metabolism indirectly through these associated pathways.

KEGG - 搜索 x KEGG COMPOUND: C00051 x K

https://www.genome.jp/entry/C00051

S-L-Glutamyl-L-cysteinylglycine;
N-(N-gamma-L-Glutamyl-L-cysteinyl)glycine;
gamma-L-L-Glutamyl-L-cysteinyl-glycine;
GSH;
Reduced glutathione

Formula C10H17N3O6S

Exact mass 307.0838

Mol weight 307.33

Structure

C00051
Mol file | KCF file | DB search

Remark Same as: D00014

Reaction R00094 R00115 R00120 R00274 R00494 R00497 R00499 R00527
R00547 R00900 R01108 R01109 R01110 R01111 R01113 R01262
R01292 R01736 R01875 R01917 R01918 R02530 R02824 R03059
R03082 R03167 R03522 R03822 R03915 R03956 R03984 R04039
R04090 R04860 R05267 R05269 R05402 R05403 R05714 R05717
R05748 R06982 R07002 R07003 R07004 R07023 R07024 R07025
R07026 R07034 R07035 R07069 R07070 R07083 R07084 R07091
R07092 R07093 R07094 R07100 R07113 R07116 R07124 R08280
R08350 R08351 R08352 R08353 R08354 R08355 R08511 R08512
R08678 R09338 R09367 R09368 R09409 R11411 R11650 R11652
» show all

Pathway map00270 Cysteine and methionine metabolism
map00480 Glutathione metabolism
map01100 Metabolic pathways
map01240 Biosynthesis of cofactors
map02010 ABC transporters
map04216 Ferroptosis
map04918 Thyroid hormone synthesis
map04976 Bile secretion
map05208 Chemical carcinogenesis - reactive oxygen species
map05415 Diabetic cardiomyopathy

Fig. R12 Glutathione metabolism relativity pathways by KEGG web retrieve.

Glutathione metabolism is closely linked to pathways such as cysteine and methionine metabolism, biosynthesis of cofactors, ABC transporters, and thyroid hormone synthesis by KEGG web retrieve. And all these pathways were prominently represented and significantly suppressed following PB@MCs treatment..

Further analysis of the original KEGG enrichment data (Figure S13, with pathways ranked by ascending P-values) revealed that the glutathione metabolism pathway exhibited a statistically significant P-value < 0.05 and a Differential Abundance Score (DA Score) of -0.7143. These results confirm that glutathione metabolism is a differentially regulated pathway and was significantly inhibited by PB@MC treatment. Corresponding revisions have been incorporated into Figures 4F and S14, and the KEGG enrichment data have been included in the “Source data”.

14. Lines 243-246: Background information on DAMPS and DC maturation should have citations.

Response: We thank the reviewer's suggestion. The background information on DAMPS and DC maturation have citations in revised manuscript.

15. Figure 5D: It should be clarified in text or Figure legend that DC maturation was examined *in vitro*.

Response: We thank the reviewer's suggestion. We have clarified that DC maturation was examined *in vitro* in revised manuscript.

16. Line 265: Figure 5E shows an increase in CD8 T cells but does not assess their cytotoxic potential, recommend removing the word “cytotoxic.”

Response: We agree with the reviewer's suggestion. We have removed the word in revised manuscript.

17. Line 281: From the results presented in Figure 6, it appears all tumors treated with PB@MCs resolved completely, but the stated results say “almost total tumour eradication.” If there was a tumor that did not resolve, this data should be included.

Response: We thank the reviewer's suggestion. We have reworded the sentence in lines 326-328 of the revised manuscript, as “Significantly, the volume metrics substantiated that PB@MCs provided the most pronounced antitumor response, leading to completely tumour eradication.”

18. Line 294: TUENL should be TUNEL.

Response: We thank the reviewer's comment. We have corrected the word in revised manuscript.

19. Lines 298-299: The data presented shows a reduction in CD206 staining, which could be an overall reduction in macrophage infiltration or a specific reduction in M2-like macrophages. The data shown does not show differentiation to an M1-like phenotype. This is again stated in the discussion (Line 375) and Figure 8, however, the conclusion that PB@MCs induce a transition from M2 to M1 phenotypes is not supported by the data shown.

Response: We thank the reviewer's comment. We agree with the reviewer that the data presented can only explain that PB@MCs caused reduced M2-like macrophages infiltration. We have reworded the sentence in Line 349-353. As “PB@MCs treatment significantly ameliorated CD4⁺ and CD8⁺ production, enhanced CD3 expression, and reduced CD206 expression in tumour sections, indicating the activation of T cells and the reduced M2-like macrophages infiltration within the tumour microenvironment.” and the sentence in Line 430-432. As “Moreover, increase of IFN- γ ; increase ratio of CD4⁺ T cells, CD8⁺ T cells and mature dendritic cell; and the reduced M2-like

macrophages infiltration in tumour sections” The changes have been highlighted in an attached file, “Manuscript with highlighted changes”.

20. Lines 320-321: “H&E...metastases” This is an incomplete sentence that reads more like a figure legend heading, suggest rewording.

Response: We thank the reviewer's suggestion. We have reworded the sentence in lines 375 of the revised manuscript, as “H&E staining was performed to evaluate the histopathological features of hepatocarcinoma tumors and their pulmonary metastases.”

21. Line 333: Recommend removing “Or” at the start of the sentence.

Response: We thank the reviewer's suggestion. We have deleted the “Or” at the start of the sentence.

22. Line 363: inhibiting should be inhibited

Response: We thank the reviewer's comment. We have corrected the word in revised manuscript.

23. Line 374: From the data shown, the differentiation of CD4 and CD8 T cells was not assessed. This data should be presented or the sentence should be reworded to avoid confusion.

Response: We thank the reviewer's suggestion. We have reworded the sentence in lines 430 of the revised manuscript to avoid confusion, as “Moreover; increase of IFN- γ ; increase ratio of CD8⁺ T cells and mature dendritic cells” The changes have been highlighted in an attached file, “Manuscript with highlighted changes”.

24. Line 400: Resolution of hepatocarcinoma tumors in rabbits was not resolved in 2 days by the data shown. Recommend rewording sentence for better clarity.

Response: We agree with the reviewer's comment. We have reworded the sentence in lines 456 of the revised manuscript to avoid confusion, as "A single injection of PB@MC is required to effectively eradicate large tumors, including melanoma and hepatocarcinoma tumours in mice and rabbits."

25. Lines 544-561. In the methods "Cell Culture" section details on how cells were cultured is included twice with almost identical wording. This duplication should be fixed.

Response: We thank the reviewer's suggestion. We have deleted the repeated sentence in revised manuscript to avoid confusion.

26. Lines 598-607. The methods description of how ROS were detected *in vivo* should clarify when and how mice were treated and at what time post-treatment tumors were harvested.

Response: We thank the reviewer's suggestion. As the same question asked by reviewer 1.4. We adopted an alternative *in vivo* method for detecting ROS. Since Neutral Red is a Type I photosensitizer (PS) generating $\bullet\text{OH}$, we selected a $\bullet\text{OH}$ assay kit (Tissue $\bullet\text{OH}$ Assay Kit, BBoxProbei O28, BB-46072, BestBio, Shanghai, China) (*Theranostics* 11, 379-396 (2021); *Chemical Engineering Journal* 411, (2021); *Biomaterials* 249, 120054 (2020)), to evaluate the $\bullet\text{OH}$ levels produced by PB@MCs in mouse tumor tissues *in vivo*. Four groups of B16 melanoma-bearing mice were injected intratumorally with 50 μL PBS (control), B@MCs, P@MCs, or PB@MCs. After 10 hours, tumors were aseptically collected, weighed, and processed according to the manufacturer's instructions. Tumor tissues were homogenized by cryo-grinding at

60 Hz for 120 s, then adjusted to 50 mg/mL with sterile PBS. Subsequently, 190 μ L of tumor homogenate (50 mg/mL) and 10 μ L of BBoxiProbe O28 working solution were added to a 96-well black plate and incubated at 37°C in the dark for 20 min. Fluorescence intensity was measured using a microplate reader (excitation: 488 nm; emission: 520 nm). Protein concentrations in the homogenate were quantified using the Bradford Protein Assay Kit (P0006, Beyotime Biotechnology, Shanghai, China). Tissue \bullet OH levels were expressed as fluorescence intensity normalized by protein concentration. As shown in **Figure R2**, mice treated with PB@MCs exhibited significantly higher fluorescence intensity compared to the other groups, demonstrating enhanced ROS (\bullet OH) generation and confirming the *in vitro* findings.

Fig. R2 Characterisation of ROS generation efficiency *In Vivo* by PB@MCs.

In vivo detection of ROS content. ROS production was assessed using the Tissue hydroxyl radicals Assay Kit (BBoxiProbe O28 probe). For comparison, B16 tumor-bearing mice of either treated by 50 μ L PBS (control group), Empty MCs linked to photosensitizer (P@MCs, contain 0.055 μ mol NR), MCs encapsulating BB170 bacteria (B@MCs), or PB@MCs for 10 h. 190 μ L of B16 tumor homogenate (50 mg/mL) and 10 μ L of BBoxiProbe O28 working solution were added to a 96 black well plate, incubated at 37°C in darkness for 20 minutes, and fluorescence intensity was detected by a Microplate Reader (excitation at 488 nm, emission at 520 nm) (n = 3 biologically independent samples). The data are presented as mean \pm SD. Statistical significance is noted with ****p < 0.0001, compared to the data for P@MCs, B@MCs and control

group according to one-way ANOVA test combination with Tukey's multiple comparisons by GraphPad Prism 9 XML project software.

We have made the corresponding modifications in lines 211-216 of revised manuscript, as “Following the experiment, tumors were aseptically harvested, weighed, and processed according to the Tissue •OH assay kit (BBoxiProbe O28 probes) to evaluate the *in vivo* •OH levels. As shown in **Fig. 3B**, mice treated with PB@MCs exhibited significantly higher fluorescence intensity compared to the other groups, demonstrating enhanced •OH generation and confirming the *in vitro* findings.” and in lines 661-673 as “To evaluate the •OH levels produced by PB@MCs in mouse tumor tissues *in vivo*. Four groups of B16 melanoma-bearing mice were injected intratumorally with 50 μ L PBS (control), B@MCs, P@MCs, or PB@MCs. After 10 hours, tumors were aseptically collected, weighed, and processed according to the manufacturer's instructions. Tumor tissues were homogenized by cryo-grinding at 60 Hz for 120 s, then adjusted to 50 mg/mL with sterile PBS. Subsequently, 190 μ L of tumor homogenate (50 mg/mL) and 10 μ L of BBoxiProbe O28 working solution were added to a 96-well black plate and incubated at 37°C in the dark for 20 min. Fluorescence intensity was measured using a microplate reader (excitation: 488 nm; emission: 520 nm). Protein concentrations in the homogenate were quantified using the Bradford Protein Assay Kit (P0006, Beyotime Biotechnology, Shanghai, China). Tissue •OH levels were expressed as fluorescence intensity normalized by protein concentration.” All changes have highlighted the revised portions.

27. Lines 699-702: Immunostaining for CD4, CD8, CD206, and CD86 was only shown in mouse tissues, the text here implies this was also performed with rabbit tumors.

Response: We agree with the reviewer's comment. We have reworded the sentence in lines 771-773 of the revised manuscript to avoid confusion, as “Rabbit tissues were processed for H&E staining and immunostaining for TUNEL and CD3 markers, whereas mouse tissues were subjected to immunostaining for TUENL, CD4, CD8, CD3, and CD206 markers.”

28. Figure 2 Legend: The legend should clarify what the colors shown in 2B represent (ie green is live cells and red is dead). (E) Legends stats PI staining is shown, but text and figure itself have DAPI staining. This should be clarified.

Response: We thank the reviewer's suggestion. We added more clarifications according to the reviewer's comments in figure legend of Figure 2B in revised manuscript as "Fluorescence microscopy images following LIVE/DEAD staining(Calcein/PI Assay Kit). HepG3B and VX2 cells were treated with 200 μ L of either PBS, Empty MCs linked to photosensitizer (P@MCs), MCs encapsulating BB170 bacteria (B@MCs), or PB@MCs for 8 h, followed by staining with a LIVE/DEAD kit for microscopy visualisation. Red fluorescence represents dead cells and green fluorescence represents live cells." In Figure 2 E, Annexin V-FITC/DAPI Apoptosis Kit (Elabscience, E-CK-A252) were used in this experiment. We have corrected and clarified in figure legend of Figure 2E in revised manuscript as "cells stained with Annexin V-FITC(+) and DAPI(-) are defined as early apoptotic cells, while cells stained with Annexin-FITC(+) and DAPI(+) are defined as late apoptotic cells for analysis by flow cytometry."

29. Figure 4: The title suggests the metabolomics of the PB@MCs themselves are being investigated, when the data actually examines how PB@MCs alter metabolism of tumor cells. This should be clarified. (D) By what criteria were the top 30 metabolites selected: p-value, fold-change, or another metric?

Response: We thank the reviewer's suggestion. We added new title to clarifications according to the reviewer's comments in line 233 as "Metabolism analysis of tumor with PB@MCs treatment" in revised manuscript.

The top 30 metabolites we selected based on relative abundance. Figure 4D shows the normalized metabolite expression levels of these top 30 most abundant metabolites (*Nat Commun.* 2025, 16(1):1347; *Nat Commun.* 2023, 14(1):2485; *ACS Nano.* 2023,

17(17):16396-16411). This analysis highlights significant differences in metabolite expression between the PB@MC treatment group and control groups. Additional clarification has been added to the manuscript in line 252: “We further performed cluster analysis of the metabolites derived from metabolic profiling. As shown in Figure 4D, metabolite expression levels from tumor tissues under different treatments were standardized using Z-score normalization. The top 30 metabolites ranked by relative abundance were selected for the clustered heatmap. The results demonstrate significant differences in metabolite expression between the PB@MC-treated group and the other groups.

30. Figure 5 Legend: For panels D-E, it should be specified what the proportion of cells shown are out of (ie % mature DCs out of total DCs, % CD4 of CD45+). Panel F, y-axis includes (%) but a ratio is not a percentage.

Response: We thank the reviewer's suggestion. For panels D-F in Figure 5 and Figure 5 legend were changed in revised manuscript. We also made additional flow cytometry gating strategy (**Figure R13**) into supplementary materials as SI Fig. 23.

Fig. R13 Flow cytometry gating strategy for the analysis of DC in vitro and T cells in the tumor.

31. Figure 6H: It should be clarified what FITC staining represents.

Response: We thank the reviewer's suggestion. During apoptosis, chromosomal DNA undergoes strand breaks, producing numerous free 3'-OH termini. These termini can be fluorescently labeled with FITC-conjugated deoxyuridine triphosphate (dUTP-FITC) *via* terminal deoxynucleotidyl transferase (TdT). As normal or proliferating cells typically show minimal DNA fragmentation and lack exposed 3'-OH groups, they exhibit negligible staining. We have incorporated these clarifications into the revised manuscript.

32. Figure 7H: It should be clarified what is included in the merge image. Is the blue staining DAPI? (I) Is the outline in the liver panels indicating tumor tissue? Please clarify.

Response: Thanks for asking this question. In Figure 7H, CD3⁺ immunofluorescence and TUNEL staining were performed on tumor tissues using a CD3⁺ Assay Kit (Cat. 552774) and a TUNEL Assay Kit (Servicebio technology CO., Ltd., Wuhan, China, G1502), respectively. Cell nuclei were stained blue using DAPI under UV excitation, whereas positive signals appeared as red fluorescence due to labeling with anti-CD3-PerCP-Cy5.5 (CD3⁺ immunofluorescence) or Cy3-dUTP (TUNEL staining). In Figure 7I, the outlined areas in the liver panels indicate tumor tissues. These clarifications have been incorporated into the revised figure legends.

Reviewer #4 (ECR):

Response: Thanks for the thorough review of our manuscript. To improve this study, we performed more experiments and added more clarifications according to all reviewers' comments.

Reviewer #1(Remarks to the Author):

1. Give examples of other papers where metabolomics analysis has been used to determine mechanisms of PDT.

Response: Thanks the reviewer's suggestion. The examples of other papers where metabolomics analysis has been used to determine mechanisms of PDT have citations in revised manuscript.

1. Ahmad, F. et al. Codoping Enhanced Radioluminescence of Nanoscintillators for X-ray-Activated Synergistic Cancer Therapy and Prognosis Using Metabolomics. *ACS Nano* 13, 9, 10419-10433 (2019).

2. Machuca, A. et al. Integration of Transcriptomics and Metabolomics to Reveal the Molecular Mechanisms Underlying Rhodium Nanoparticles-Based Photodynamic Cancer Therapy. *Pharmaceutics* 13, 1629 (2021).

2. They used "bioluminescent bacteria" in title as requested but then continued to use "bioluminous bacteria" in the remainder. Please correct this throughout

Response: Thanks for the reviewer's suggestion. We have carefully go through the full manuscript and used "bioluminescent bacteria". The changes have been highlighted in revised manuscript.

3. "The generation of OH is independent of oxygen availability" is incorrect, Should be "The generation of OH is less dependent on oxygen availability"

Response: We thank the reviewer's suggestion. We have corrected these words in revised manuscript.

4. Change "via an electrostatic field" to "via electrostatic attraction"

Response: We thank the reviewer's suggestion. We have corrected these words in revised manuscript.

5. What does "surface opaque melanoma" mean?

Response: We thank the reviewer's suggestion. It is mice melanoma model. We have corrected these words in revised manuscript.